# De novo identification of microbial contaminants in low microbial biomass microbiomes with Squeegee

Yunxi Liu[1], R. A. Leo Elworth[1], Michael D. Jochum[2], Kjersti M. Aagaard [2] & Todd J. Treangen [1] ✉

Computational analysis of host-associated microbiomes has opened the door to numerous discoveries relevant to human health and disease. However, contaminant sequences in metagenomic samples can potentially impact the interpretation of findings reported in microbiome studies, especially in low-biomass environments. Contamination from DNA extraction kits or sampling lab environments leaves taxonomic "bread crumbs" across multiple distinct sample types. Here we describe Squeegee, a de novo contamination detection tool that is based upon this principle, allowing the detection of microbial contaminants when negative controls are unavailable. On the low-biomass samples, we compare Squeegee predictions to experimental negative control data and show that Squeegee accurately recovers putative contaminants. We analyze samples of varying biomass from the Human Microbiome Project and identify likely, previously unreported kit contamination. Collectively, our results highlight that Squeegee can identify microbial contaminants with high precision and thus represents a computational approach for contaminant detection when negative controls are unavailable.

In recent years, the field of metagenomics has grown at a fast pace thanks to next-generation sequencing technologies. The scale and complexity of metagenomics studies have expanded alongside the volume of the sequencing data. By performing metagenomic sequencing, we are able to analyze the DNA and RNA of the entire microbial community in varying and heterogeneous biomass environments, such as samples from wastewater, soil, or human body sites[1]. One commonly used method is 16S rRNA gene sequencing. The 16S rRNA gene is highly conserved in bacteria and can be amplified and used as a marker gene for taxonomic classification[2–7]. The other widely used technique is whole-genome shotgun sequencing, where all DNA sequences in the community are fragmented and sequenced[2,3,8–10]. Both methods open the door for identifying members of microbial communities from the sampled environments and estimating the relative abundance of each member[1]. However, the results from both of these methods can be affected by microbial contamination. Microbial contamination occurs when sequences from microbes appear in the data that were not in the original samples[3,11].

A variety of sources can introduce microbial contamination. External sources include personnel, the laboratory environment, and kits and reagents used for collecting and processing samples[2,3,11–20]. Internal sources of contamination may include human error, such as sample mislabeling or inadvertent mixing[3,11,17,21]. Contaminant sequences have also made their way into public reference databases[22–25]. Studies have shown that contaminants in DNA extraction kits are ubiquitous[26,27], and can bear an impact on metagenomic studies, especially for low-biomass environments if they are not accounted for in the analysis[11,20,28]. For example, in a recent nasopharyngeal microbiota study on newborn babies conducted in Thailand, contaminants found in DNA extraction kits resulted in contaminant bias[3].

[1]Rice University, Department of Computer Science, Houston, TX 77005, USA. [2]Department of Obstetrics and Gynecology, Division of Maternal-Fetal Medicine, Baylor College of Medicine and Texas Children's Hospital, Houston, TX 77030, USA. ✉e-mail: treangen@rice.edu

Extra precautions during sample collection and processing, and well-designed experiments, such as processing samples in a clean, well-structured environment, or using depletion methods to remove host DNA, can help minimize the impact caused by contamination[11,29]. In addition, computational models have been used to identify and remove contaminants from sequenced datasets. For example, the recently published software Recentrifuge uses a score-oriented comparative approach to identify and remove contaminants from sequencing reads[30]. As is the case with all current computational methods for microbial contaminant detection, performing contamination removal with Recentrifuge requires experimental controls. Another statistical tool for identifying and removing contamination is Decontam[3]. Decontam includes a combination of a frequency-based approach and a prevalence-based approach. Auxiliary DNA quantitation data are required to perform the frequency-based analysis, and standard negative control samples are required to perform the prevalence-based analysis[3].

Experimental negative and/or environmental contaminant controls combined with computational contamination identification and removal is effective[3,19,30]. However, the additional costs (both time and resources) to include negative control experiments are often a barrier to utilization. As a result, negative control experiments available for publicly available datasets are often lacking. Although contaminant sequences have been a known issue for some time, negative control data are often unavailable in public databases, making it nearly impossible to perform contamination removal on uploaded data.

Since the composition of contaminants within DNA extraction kits and other lab reagents are ubiquitous and can be distinct, our hypothesis is that contaminants from the same sources, such as DNA extraction kits or from a lab environment, will share similar characteristics in the composition of their contaminants. This fact should enable contaminants to be found in the form of shared species in samples taken from sufficiently distinct ecological niches, or in our case, body sites. In particular, this proposed approach is most relevant when the sequencing runs use the same DNA extraction kit and/or are processed in the same lab after reaching sufficient sequencing depth.

## Results

### Overview of our experimental evaluation of Squeegee

In this work, we have implemented a de novo computational contamination detection tool, Squeegee, which is able to identify potential contaminants at the species level. Squeegee performs taxonomic classification and searches for shared organisms across multiple samples and sample types. The workflow of the pipeline is shown in Fig. 1.

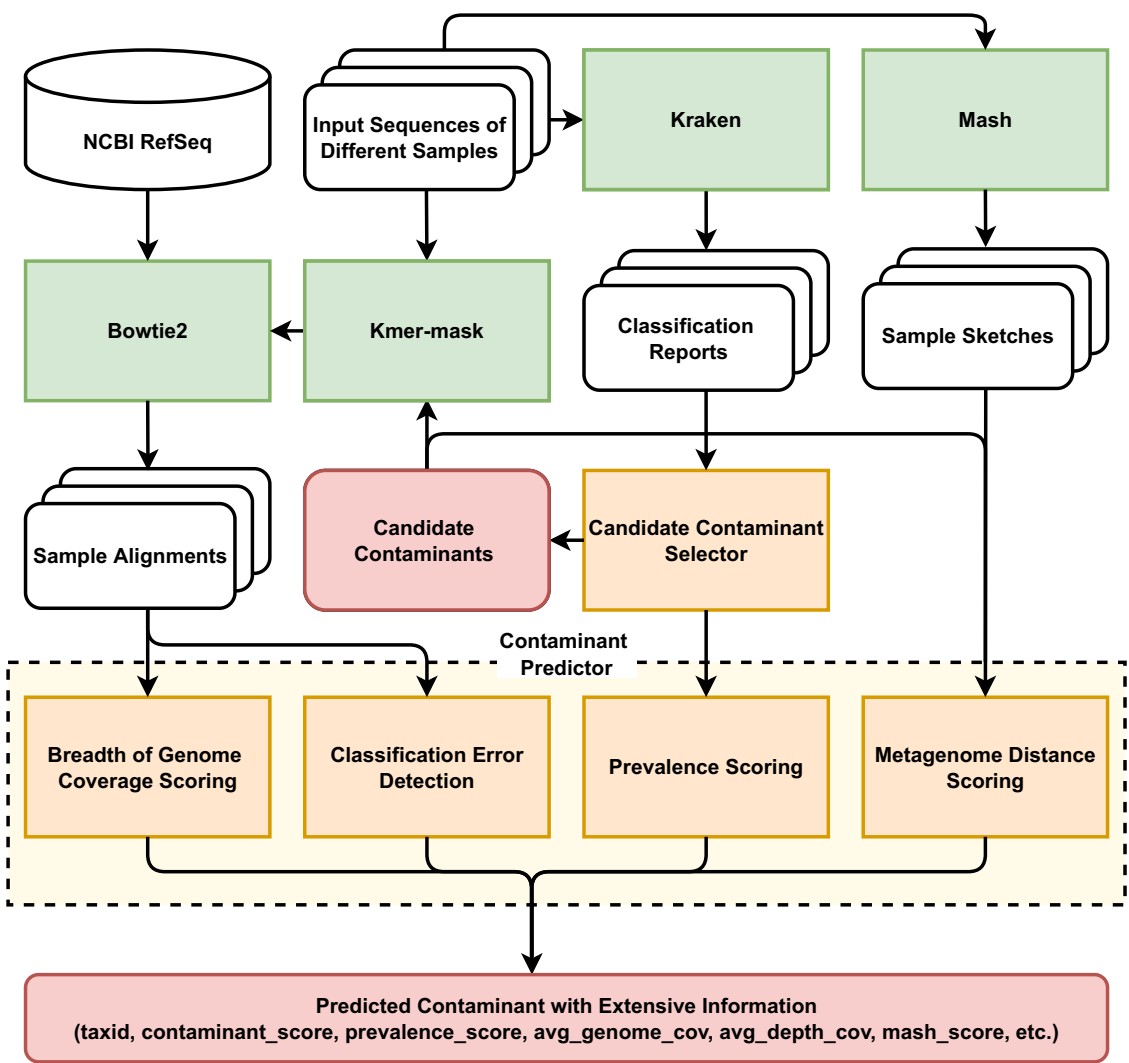

**Fig. 1 | Squeegee pipeline workflow.** Squeegee starts with taxonomic classification using Kraken to determine a set of candidate contaminant species. Reads from the input data are aligned to the representative genomes of the candidate contaminant species using Bowtie2 in multi-alignment mode. It also calculates the pairwise Mash distance for all the samples. Then, it combines the prevalence score, the Mash distance, as well as the breadth/depth of genome coverage of the candidates to predict potential contaminants.

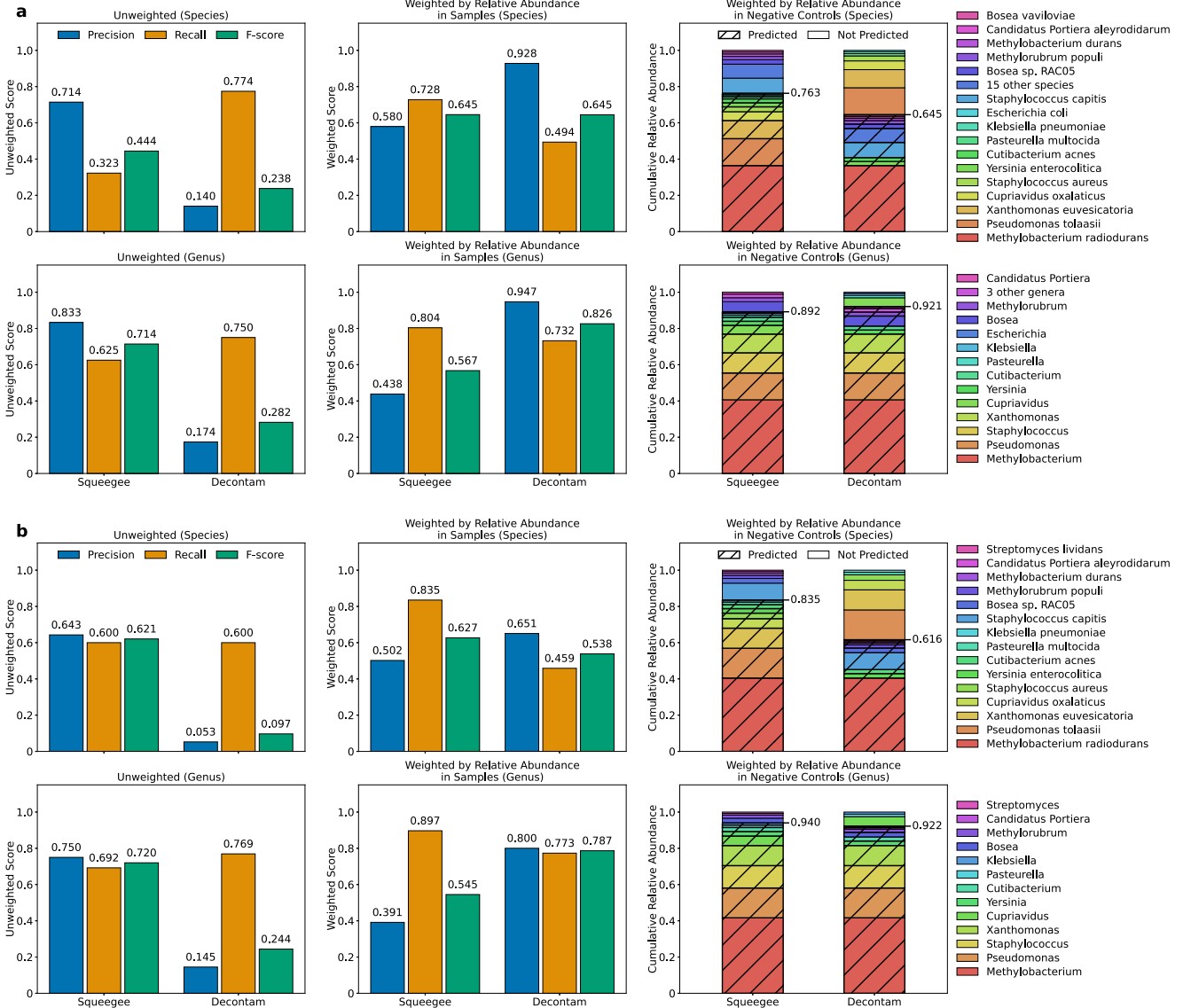

**Fig. 2 | Benchmarking Squeegee with Decontam on the maternal/infant dataset.** Squeegee (de novo) and Decontam (with negative control) accuracy at species and genus ranks are evaluated with (**a**) the permissive ground truth and (**b**) the more strict ground truth. The figures show the precision, recall, and *F*-score calculated at species and genus rank for both methods. The unweighted precision is calculated as the ratio between the number of predicted contaminant taxa found in the ground truth and the total number of predicted contaminant taxa. The unweighted recall is calculated as the ratio between the number of predicted contaminant taxa found in the ground truth and the total number of taxa in the ground truth. While weighted by samples, the measurements are weighted by the mean proportion of the reads assigned to each taxon in the non-control experiment samples. The weighted by negative controls figures show the detailed composition of the taxa, their mean relative abundance in the negative control samples, and the cumulative relative abundance of the correctly predicted putative contaminants (weighted recall) by different methods. The correctly predicted species/genera are marked with strips, and the species/genera that the methods failed to predict are without stripes. Multiple low relative abundance taxa have been combined in **a**. Source data are provided as a Source Data file.

The software takes multiple samples containing sequencing data collected from distinct microbiomes as input and then uses taxonomic classification to search for candidate contaminant species that are shared across samples. By estimating pairwise similarity between metagenomic samples that the candidate contaminant species presents, and calculating breadth and depth of genome coverage by aligning the reads to the reference genome of the candidate contaminant species, Squeegee identifies taxonomic classification errors and makes accurate contaminant predictions at the species rank by filtering false calls from the candidates.

We evaluated Squeegee on three datasets, including (i) a simulated dataset with ground truth contaminants, (ii) a real dataset with negative controls, and (iii) HMP samples without negative controls but with associated DNA extraction kit contaminants. Details on the implementation and evaluation of Squeegee can be found in the methods section. The dataset characteristics and parameters used in the study can be found in Supplementary table 1.

## Stable community members for human body sites
In order to accurately identify contaminant sequences from external sources, such as lab environments or reagents used during the extraction or sequencing process, stable community members from different sample types must be considered. To assess whether there are ubiquitous genera across body sites comprising the human microbiome, we identified the stable community members across different human niches using Kraken classification results for HMP samples (Supplementary Table 2). By looking at each set of common community members of different body sites, we found no genera

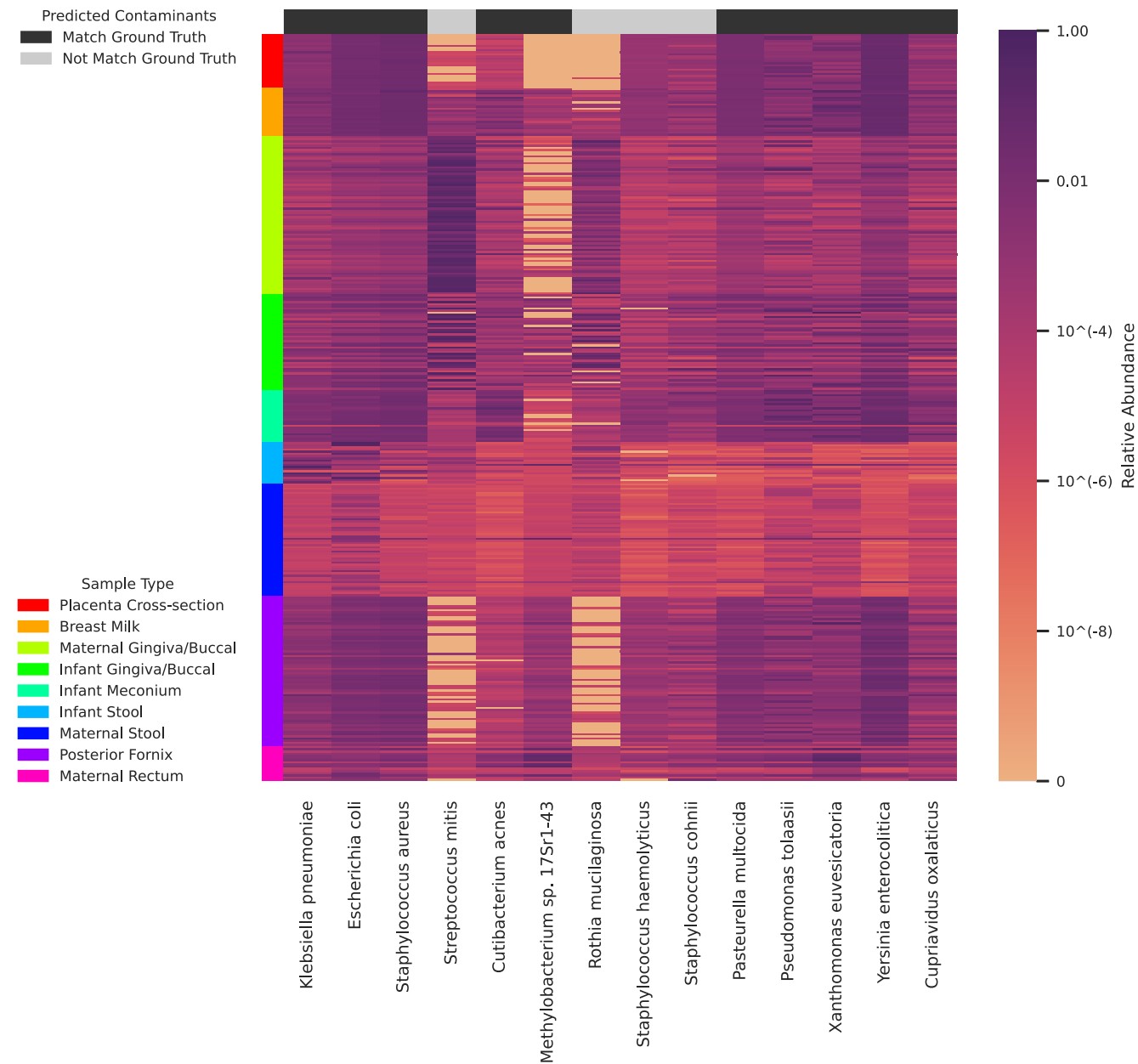

**Fig. 3 | Relative abundance of all predicted species in the maternal/infant dataset.** The samples are clustered by their sample type, which is shown with different colors on the color label on the y-axis. The predicted contaminant species that can be found in the permissive ground truth contaminants are marked by the black label on the x-axis, whereas the predicted contaminant species that do not match the strict ground truth contaminants sample are marked in gray. Source data are provided as a Source Data file.

present in more than three of the six body sites (oral, nasal, skin, stool, throat, and vaginal).

## Benchmark with Decontam

We evaluated Squeegee prediction accuracy at both genus and species rank on the maternal/infant datasets. During this benchmark, Squeegee performed contamination prediction without using the negative control samples, while Decontam took the classification results of the 10 negative control samples as input for contamination identification. A permissive ground truth contaminant set, and a strict version of the ground truth contaminant set, are generated with data from the negative control samples as well to use as a reference for the evaluation, whereas the strict set is generated with more stringent filtering to ensure high confidence. The details of the contaminant ground truth sets can be found in the methods section.

Figure 2a shows the precision, recall, and F-score of Squeegee and Decontam at both species and genus rank using the permissive ground truth set. The unweighted precision, unweighted recall, and unweighted F-score for Squeegee are 0.714 (10/14 species), 0.323 (10/31 species), and 0.444 at species rank, and 0.833 (10/12 genera), 0.625 (10/16 genera), and 0.714 at genus rank, respectively. The false positive calls for Squeegee are *Rothia mucilaginosa*, *Staphylococcus cohnii*, *Staphylococcus haemolyticus*, and *Streptococcus mitis*. The unweighted precision, unweighted recall, and unweighted F-score for Decontam are 0.140, 0.774, and 0.238 at species rank, along with 0.174, 0.750, and 0.282 at genus rank, respectively.

We also evaluated both methods with weighted scores, taking into account the abundance of information. Each of the species are first weighted by the mean fraction of reads assigned to those species in the non-negative samples. The weighted precision, weighted recall, and

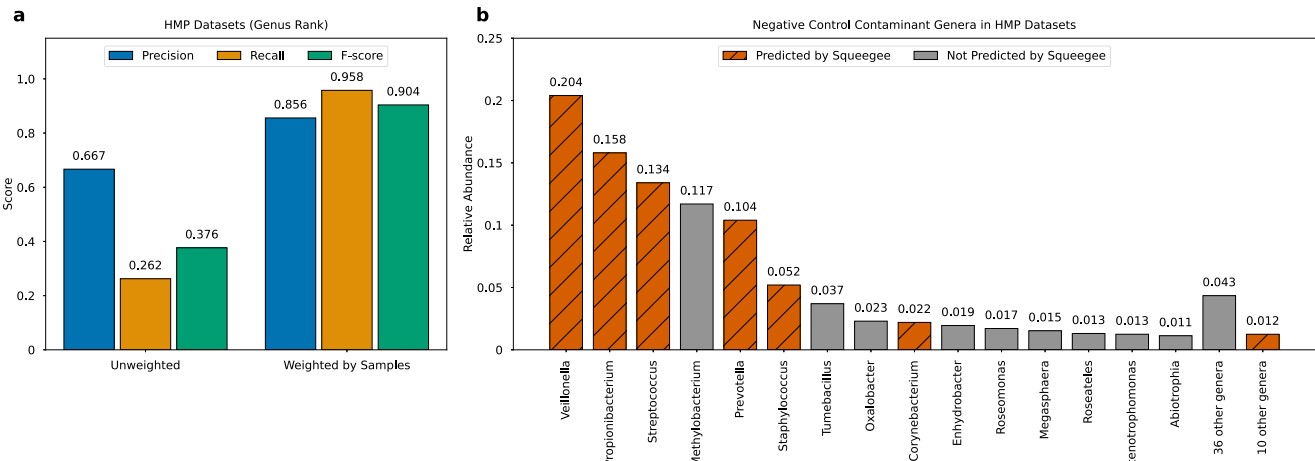

**Fig. 4 | Squeegee performance on HMP metagenomic datasets. a** Left panel depicts the Genus level precision, recall, and *F*-score using previously reported kit contaminants as the ground truth. Unweighted precision is calculated as the ratio between the number of predicted contaminant taxa found in the ground truth and the total number of predicted contaminant taxa. An unweighted recall is calculated as the ratio between the number of predicted contaminant taxa found in the ground truth and the total number of taxa in the ground truth. While weighted by samples, the measurements are weighted by the mean proportion of the reads assigned to each taxon in the non-control experiment samples. **b** The right panel highlights the correctly predicted genera marked in orange with stripes, and the genera that Squeegee failed to predict are marked in gray. Genera with relative abundance below 1% are combined. Source data are provided as a Source Data file.

weighted *F*-score for Squeegee were 0.580, 0.728, and 0.645, respectively, and for Decontam were 0.928, 0.494, and 0.645, respectively, at species rank. The same measurements at genus rank were 0.438, 0.804, and 0.567 for Squeegee, respectively, and 0.947, 0.732, and 0.826 for Decontam, respectively.

More importantly, we took a closer look at the predicted contaminants output by each method and evaluated the recall weighted by the relative abundance of the taxa in the negative control samples. Although Squeegee failed to identify several putative low-abundance contaminant species, the 10 correctly predicted species by Squeegee occupy over 0.763 of the cumulative relative abundance from the composition of the putative ground truth contaminants. With the same measurement, the species rank weighted recall under the same criteria for Decontam is 0.645. At genus rank, both methods performed well, with weighted recall for Squeegee scored at 0.892 and Decontam scored at 0.921. Although Decontam mislabeled some of the high abundance contaminants at species rank, it did label some of the closely related species under the same genera as contaminants, resulting in a significant increase of the score at genus rank. Figure 2b shows the accuracy of Squeegee and Decontam using the strict ground truth set. The detailed results can be found in Supplementary Section 1.

Figure 3 shows the relative abundance of all contaminant species predicted by Squeegee in each of the non-control samples. The samples are clustered by sample types, designated by their color label on the y-axis. The predicted contaminant species that can be found in the permissive ground truth contaminant set are labeled in black at the top of the figure, and the predicted contaminant species not found in the permissive ground truth contaminant set are labeled in light gray.

### Evaluation of Squeegee on the HMP datasets

We evaluated Squeegee prediction accuracy on the HMP datasets as well. Figure 4 shows the precision, recall, and *F*-score of Squeegee predictions at genus rank. Squeegee has an unweighted precision of 0.667 (16/24 genera), an unweighted recall of 0.262 (16/61 genera), and an unweighted F-score of 0.376. While each taxa is weighted by their relative abundance from the non-control samples, Squeegee achieved a weighted precision of 0.856, a weighted recall of 0.958, and resulted in a weighted *F*-score of 0.904. Figure 4 also shows the relative abundance of true contaminant genera identified in the MoBio DNA extraction kit. The contaminants successfully predicted by Squeegee are colored orange with stripes, and the contaminants Squeegee failed

to predict are colored gray. Low-abundance genera with relative abundance below 1% are combined in the figure. Although only 16 genera were correctly predicted, those genera accounted for the majority of the contaminated reads in the ground truth with a total relative abundance of 0.686.

Since we use bacteria identified at the genera level as inherent putative contaminants in the MoBio DNA extraction kit level[18] for our negative control reference, accuracy measurements at the species level do not apply. It is worth noting that more than 81.3% (61 out of 75) of species Squeegee predicted as contaminant species in the HMP datasets fell under the ground truth contaminant genera. Supplementary Fig. 1 shows the prevalence, the breadth of genome coverage, and additional score and filtering information of the top 50 predicted contaminant species after filtering. The first 16 rows show the prevalence of each species among each of the sample types, where zero prevalence is marked in blue. The following 16 rows show the breadth of genome coverage of each species in each sample type. The remaining rows show the prevalence score, the alignment score, the Mash score, and the combined score used to make the final prediction and whether each species passes the filters. The last row of the heat map shows whether the species can be found in the ground truth, with true positives shown in white and false positives shown in black. Detailed information on all candidate contaminant species can be found in Supplementary Fig. 2.

### Evaluation of Squeegee on simulated datasets

To test the contamination limit of detection of Squeegee, we designed a set of simulated datasets based on the taxonomy profile of the real-world metagenomic samples. About 126 samples are simulated and divided into three groups based on the different relative abundance of the spike-in contaminant sequences (0.25, 0.50, and 1.00%). There are 42 simulated datasets in each of the groups, representing microbial communities from seven distinct environments. The details of how those simulated datasets are generated can be found in the methods section.

Figure 5 shows the unweighted precision, recall, and F-score of different simulated sample groups at species rank and the same measurement weighted by the relative abundance of the taxa in the non-control samples. The figure also shows the detailed composition of the taxa, their relative abundance in the spike-in contaminant community, and the cumulative relative abundance of the correctly

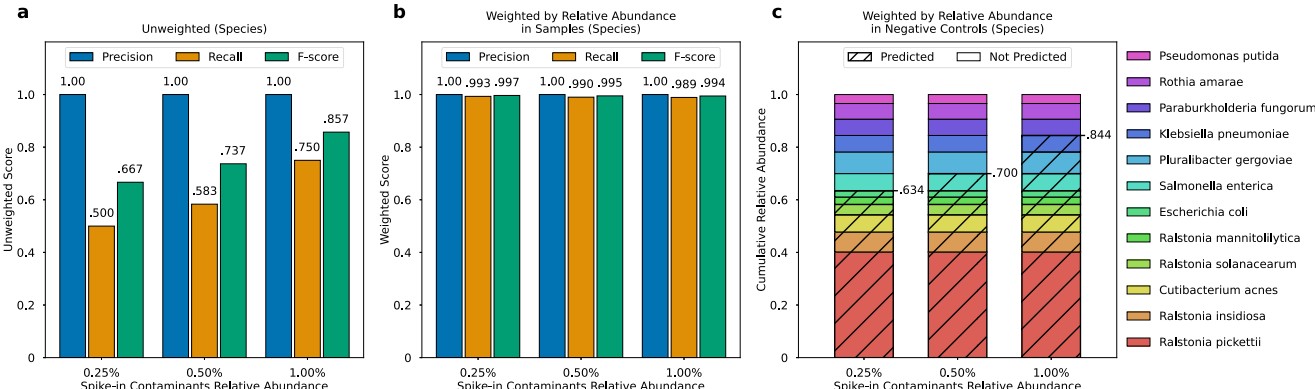

**Fig. 5 | Squeegee prediction accuracy at species ranks on the simulated datasets. a** The leftmost panel shows the precision, recall, and F-score calculated at species rank for the different relative abundance of spike-in contaminants. The unweighted precision is calculated as the ratio between the number of predicted contaminant taxa found in the ground truth and the total number of predicted contaminant taxa. The unweighted recall is calculated as the ratio between the number of predicted contaminant taxa found in the ground truth and the total number of taxa in the ground truth. **b** The center panel shows the same measurements weighted by the mean proportion of the reads assigned to each taxon in the non-control simulated samples. **c** The right panel shows the detailed composition of the taxa, their relative abundance in the spike-in contaminant community, and the cumulative relative abundance of the correctly predicted contaminants at a different relative abundance of spike-in. The correctly predicted species are marked with striped lines, and the species Squeegee failed to predict are without striped lines. Source data are provided as a Source Data file.

predicted contaminants at different relative abundances of spike-in. At all three different spike-in levels, where contaminant sequences occupied 0.25, 0.50, and 1.00% of the total reads, Squeegee had the perfect precision of 1.0. For unweighted recall, the 0.25% spike-in group scored 0.500, the 0.5% spike-in group scored 0.583, and the 1.0% spike-in group scored 0.750. As a result, the unweighted *F*-score for the 0.25% group, 0.50% group, and 1.00% group are 0.667, 0.737, and 0.857. When each species is weighted by their relative abundance in non-control samples, the 0.25% spike-in group scored 0.993, 0.5% spike-in group scored 0.990, and 1.0% spike-in group scored 0.989 for the weighted recall, and 0.25% spike-in group scored 0.997, 0.5% spike-in group scored 0.995, and 1.0% spike-in group scored 0.994 for the weighted *F*-score.

When each species is weighted by its relative abundance in the negative control, the cumulative relative abundance of true positive prediction for the 0.25% spike-in group is 0.634. As the spike-in level increases, at 0.5% spike-in abundance, Squeegee scored 0.700, with one additional species, *Salmonella enterica*, identified as a contaminant. The cumulative relative abundance of true positive predictions continued to increase at a 1.0% spike-in abundance level, and Squeegee scored 0.844 with two more correct contaminant species predicted. In general, the unweighted recall and the cumulative relative abundance of the true positive predictions increase as the number of spike-in contaminant sequences increases since more contaminant sequences provide a stronger signal for Squeegee to pick up on and to make definite calls with respect to contamination.

### Alpha diversity analysis with contamination removal

Figure 6a shows Shannon's diversity index and Simpson's diversity index for the maternal/infant dataset before and after contamination removal. Both diversity metrics for the samples were evaluated before the contaminant reads were removed (shown in red), after removing species confirmed by the permissive ground truth contaminants (shown in blue), and after removing all species predicted by Squeegee (shown in black). The max removal cutoff is set to 1%, which only removes species with a relative abundance of less than 1%. We observed significant decreases in Simpson's diversity index in both placental and breast milk groups and significant decreases in Shannon's diversity index in the placental group. There are also significant decreases in Shannon's diversity index in the breast milk group if we remove all predicted contaminant species, but no significant decreases are found by only removing contaminant species confirmed by the negative

control experiments. For a more strict max removal cutoff of 0.5%, we still found significant decreases in both Shannon's and Simpson's diversity index in the placental group (See Supplementary Fig. 3).

Figure 6b shows the same alpha diversity analyses performed on the HMP samples with the maximum removal cutoff set to 1%. We observed significant decreases in Shannon's diversity index values in oral and nasal samples, and a significant decrease in Simpson's diversity index in oral samples. With the max removal cutoff of 0.5%, there are significant decreases in Shannon's diversity index and Simpson's diversity index in the oral samples (See Supplementary Fig. 4).

### Reagent-specific contamination detection

We applied Squeegee to a human-derived RNA-Seq dataset from an index study that aimed to evaluate the potential for contamination arising during sequencing across 6 different sequencing centers in Europe in the GEUVADIS consortium[31,32]. The resultant generated RNA sequencing data arose from 40 sequencing runs performed at all six sequencing centers on identical sequencing platforms following extraction with the same kits on parallel samples. The prediction from Squeegee indicates that seven species (*Human gammaherpesvirus 4*, *Proteus virus Isfahan*, *Escherichia coli*, *Bacillus megaterium*, *Bacillus cereus*, *Klebsiella pneumoniae*, and *Cutibacterium acnes*) are reagent specific contaminants and can be found in the sequencing runs across different sequencing centers. (Supplementary Fig. 5). While *Human gammaherpesvirus 4* is associated with the cell line used during the sequencing, *Escherichia coli*, *Bacillus megaterium*, *Bacillus cereus*, *Klebsiella pneumoniae*, *Cutibacterium acnes* can be putative common "kit contaminants" that have been previously reported. However, *E. coli* and *K. pneumoniae* are also prevalent environmental and human commensal microbes or pathobionts.

### Squeegee run time and memory usage

We tested the performance of Squeegee with datasets of different sizes. Table 1 shows the run time and peak memory usage of Squeegee for different sizes of input. The run time of Squeegee (in CPU hours) is primarily determined by the size of the input data and is also affected by the number of potential contaminants identified based on the taxonomic classification results. Since the reads are mapped to reference genomes of each potential contaminant using Bowtie2 (with multi-alignments enabled), more contaminants would increase the CPU time for alignment and the coverage calculation process. The peak memory usage of Squeegee is mainly driven by the size of the

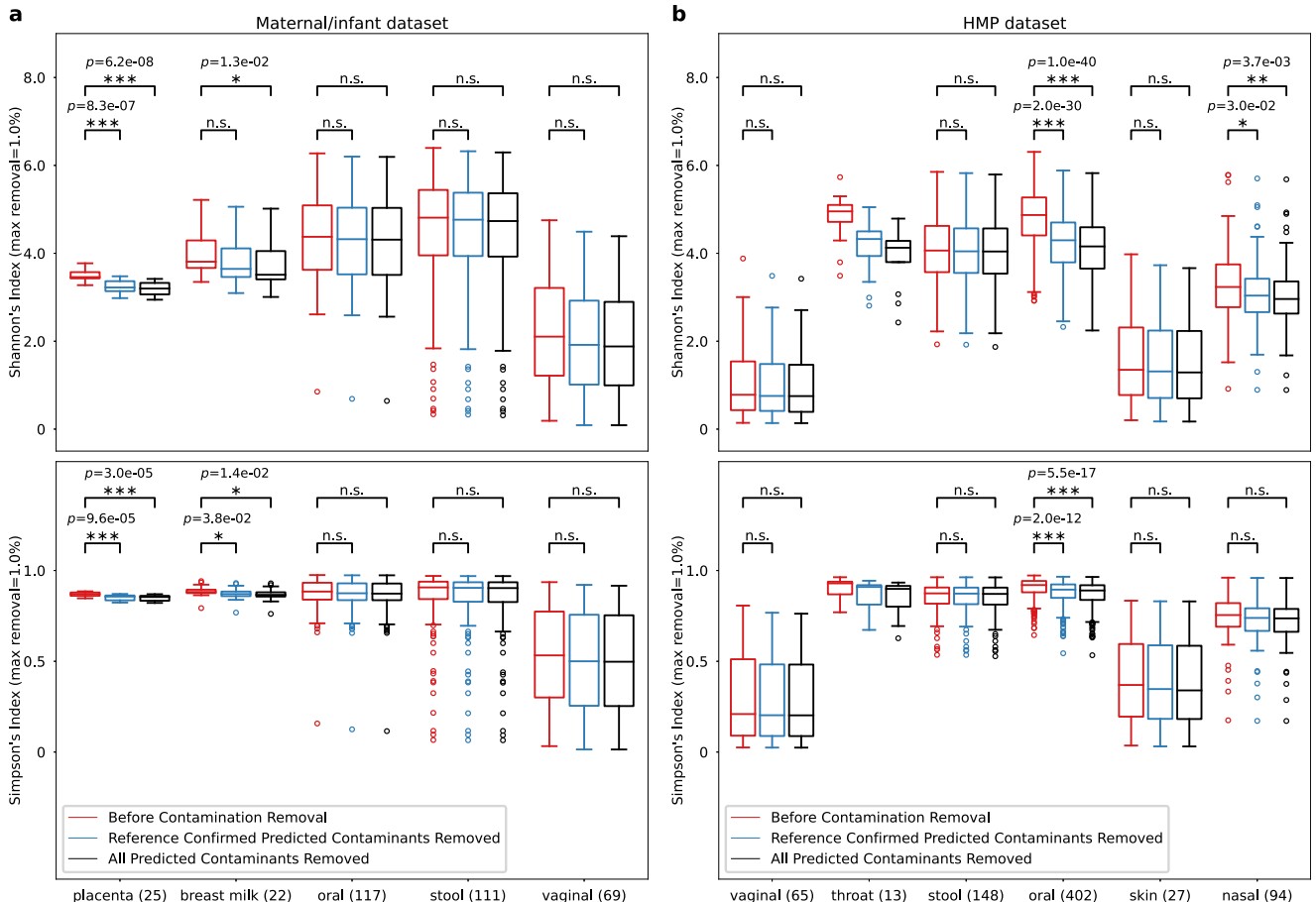

**Fig. 6 | Alpha diversity indexes before and after contamination removal.** The figure shows the alpha diversity indexes of **a** maternal/infant dataset and **b** HMP dataset. Both Shannon's and Simpson's diversity index of the communities in each of the samples were evaluated before the contaminant reads were removed (red), after removing species only confirmed by the experimental negative control (blue), and after removing all species predicted by Squeegee (black). The max removal is set to 1%. Numbers inside parentheses are the numbers of samples in each sample type. The significance test was done using a two-sided Mann–Whitney $U$–test for all combined sample types with more than 20 samples. No adjustments were made for multiple comparisons. Significance labeling: n.s.($P > 0.05$), *($P \leq 0.05$), **($P \leq 0.01$), ***($P \leq 0.001$). Each box plot includes the median line, and the box bounds the interquartile range (IQR). The Tukey-style whiskers extend from the box by at most $1.5 \times$ IQR. The circle denotes outliers that extend beyond the whiskers. In **a**, the exact $p$-value between Shannon's index before removal and reference confirmed contaminants removed is $8.3 \times 10^{-7}$ for placenta samples. The exact $p$-value

between Shannon's index before removal and all contaminants removed is $6.2 \times 10^{-8}$ for placenta samples and $1.3 \times 10^{-2}$ for breast milk samples. The exact $p$-value between Simpson's index before removal and reference confirmed contaminants removed is $9.6 \times 10^{-5}$ for placenta samples and $3.8 \times 10^{-2}$ for breast milk samples. The exact $p$-value between Simpson's index before removal and all contaminants removed is $3.0 \times 10^{-5}$ for placenta samples and $1.4 \times 10^{-2}$ for breast milk samples. In **b**, the exact $p$-value between Shannon's index before removal and reference confirmed contaminants removed is $2.0 \times 10^{-30}$ for oral samples and $3.0 \times 10^{-2}$ for nasal samples. The exact $p$-value between Shannon's index before removal and all contaminants removed is $1.0 \times 10^{-40}$ for oral samples and $3.7 \times 10^{-3}$ for nasal samples. The exact $p$-value between Simpson's index before removal and reference confirmed contaminants removed is $2.0 \times 10^{-12}$ for oral samples. The exact $p$-value between Simpson's index before removal and all contaminants removed is $5.5 \times 10^{-17}$ for oral samples. Source data are provided as a Source Data file.

database used by Kraken for taxonomic classification. The following runs use the same Kraken database (302 GB) built with NCBI RefSeq (Release 202); thus they have similar peak memory usage.

## Discussion

To the best of our knowledge, Squeegee is the first de novo computational tool specifically designed to identify and nominate taxa as potential contaminants in the absence of "kit negative", environmental

contaminant controls, and other auxiliary data. Squeegee is able to mark these taxa contained within metagenomic samples without requiring negative experimental controls, and can identify potentially widespread contaminants in publicly available data. In order to predict contaminant species, multiple pieces of evidence are taken into consideration, including the prevalence rate of species, the metagenomic distance of the samples that contain the species, and how well the genomes of those species are being covered. A recent study has shown that the accuracy of the taxonomic classification algorithm has become a limiting factor of contamination detection due to high levels of sequence similarity at species rank[33]. With a breadth of genome coverage for each contaminant species being calculated, Squeegee also attempts to address taxonomic classification error that might occur during the taxonomic binning process, which is a common issue for k-mer-based methods[34,35]. Comparisons between Squeegee predictions and experimental control data show that Squeegee is capable of accurately inferring contamination at the species level, especially in

## Table 1 | Run time and memory usage

|  | Test set 1 | Test set 2 | Test set 3 |
|---|---|---|---|
| Input size (GB) | 59.2 | 59.2 | 118.0 |
| # of potential contaminants | 15 | 18 | 26 |
| CPU hours | 55.5 | 63.3 | 130.3 |
| Peak memory usage including Kraken (GB) | 320.4 | 320.4 | 322.0 |

regard to contaminants occurring at a relative abundance of 5% or higher.

For the maternal/infant dataset, a strict contaminant ground truth and a permissive contaminant ground truth were constructed with the taxonomic assignment of the sequencing data from the negative control experiments with the use of prevalence, relative abundance, and absolute read count filtering, a common practice to minimize the taxonomic assignment error and determine the presence or absence of species[36,37], with different filtering parameters. Squeegee predicted most of the putative contaminants found in the strict ground truth contaminant set (see Fig. 2b), including species from contaminating genera (e.g., *Methylobacterium*, *Pseudomonas*, and *Xanthomonas*) that have been previously reported[2,20,38]. For the presumptive false negative contaminant species the Squeegee failed to predict, all were of relative abundances below 5% except for *Staphylococcus capitis*. In addition to other genera and species unique to the low-biomass maternal/infant samples, we also found that Squeegee predicted a number of contaminant species from the genera *Staphylococcus*, including *Staphylococcus haemolyticus* and *Staphylococcus cohnii*, that are not found in the experimental control samples. *Staphylococcus* species are often found in the normal flora of the skin and have been reported multiple times as contaminants from DNA extraction kits and laboratory environments[18,20,39]. *Staphylococcus* species are also well-known for their highly similar genomes, which creates a big challenge for the taxonomic assignment task[33]. Additionally, Squeegee identified *Rothia mucilaginosa*, which is a part of the normal oropharyngeal flora, and *Escherichia coli* as contaminants. Both species may represent bonafide species shared across body niches[40]. It is also possible that the experimental control samples were not sequenced deeply enough to reveal these species or the species were at a low enough relative abundance in the experimental control samples that were filtered out during quality control.

We benchmarked Squeegee against a "gold-standard" contamination detection approach in Decontam, with its prevalence-based method requiring negative control samples as input. We note that we view both tools as complementary, especially since using negative controls is recommended best practice for contamination removal. From the results (Fig. 2b), we see that Squeegee is able to achieve performance that meets or exceeds Decontam predictions at species rank using the strict ground truth, with respect to unweighted *F*-score, weighted *F*-score by the relative abundance in the non-control samples, and cumulative relative abundance of the putative correctly identified contaminants from the negative control experiment samples. On the other hand, at the genus rank, Squeegee is unable to match Decontam performance when *F*-score is weighted by the relative abundance in the non-control samples while performing on par with Decontam with respect to the cumulative relative abundance of the correctly identified contaminant genera in negative controls. Although Decontam failed to recognize *Pseudomonas tolaasii* and *Xanthomonas euvesicatoria* as putative contaminant species, multiple species under the same genera were successfully identified, increasing its genus rank score. While evaluating using the permissive ground truth contaminant set, Squeegee performed equally well at species rank with respect to weighted *F*-score (Fig. 2a), with a drop in unweighted recall given Squeegee failed to recognize most contaminant species with mean relative abundance less than 1% in the negative controls.

As expected, Decontam with the experimental negative control data performs best in terms of unweighted recall (see Fig. 2a). Decontam identified 24 out of 31 species within the permissive contaminant ground truth set, only missing *Pseudomonas tolaasii*, *Xanthomonas euvesicatoria*, *Cupriavidus oxalaticus*, *Staphylococcus aureus*, *Pasteurella multocida*, *Klebsiella pneumoniae*, and *Escherichia coli*. It is possible that Decontam did not flag some species as

contaminants that are shared between the source of contamination and the sampling environment, such as *Staphylococcus aureus* and *Escherichia coli*. At the same time, Squeegee identified all those seven species, which the relative abundance in the permissive ground truth adds up to 35.5%, as contaminants, which completes the entire contaminant ground truth set if we take the union of the predictions made by the two methods. Alternatively, we acknowledge that this may over-call "contamination" by virtue of shared species among body niches. This once again highlights the complementarity of Decontam with negative controls and Squeegee, and also the value of Squeegee either when negative controls are unavailable (existing metagenomic sequence datasets) or for lab contamination that affects both the negative control and samples.

In addition to identifying microbial contaminants within microbiome datasets lacking negative controls, Squeegee can also identify contaminants in human RNA-seq data. In our experiments with the GEUVADIS consortium human RNA-Seq dataset, Squeegee predicted that seven species were lab preparation related. In this dataset, since non-human reads are all classified as contamination, by identifying reagent-specific contaminants shared across different sequencing labs, one can backtrack and identify lab-specific contaminants using the classification report provided by Squeegee.

Another use case for Squeegee is to detect batch-specific contaminants, as well as cross-contaminants. Suppose negative control samples are not prepared and sequenced for every batch run. In this case, batch-specific contaminants may get mixed into the samples, causing bias in the downstream analysis. A similar scenario is that cross-contamination occurs in a single batch but happens not to affect the negative control sample. Running Squeegee on each individual batch allows the user to detect such batch-specific contaminants or cross-contaminants since the Squeegee detection method does not depend on the negative control profiles.

Squeegee is designed for de novo identification of microbial species that are likely contaminants; a higher combined contaminant score indicates the species has a higher potential for being an actual contaminant. However, Squeegee's failure to flag a microbial species in a sample as a likely contaminant does not mean it is not a contaminant. As mentioned before, one of the limiting factors is the relative abundance of the species within the source of the contamination. Figures 2 and 4 show that contaminant species with low relative abundances in the control samples are more challenging to identify since the sequencing signals of such species become even weaker in the non-control metagenomic samples. Squeegee failed to predict some of the low-abundance genera/species in the simulation dataset due to similar reasons (see Fig. 5). In order to challenge Squeegee, the simulation dataset we designed contains very low proportions (0.25–1%) of spike-in contaminant sequences. Among the 12 spike-in contaminant species, all except *Ralstonia pickettii* have relative abundance below 0.1 within the spike-in. As the relative abundance of the total spike-in sequences increases, we observed that the unweighted recall increased as well, and Squeegee is able to pick up more and more contaminant species. As shown in our experimental results (both simulated and real), Squeegee can exhibit low recall on low-abundance contaminant species, which means there will be residual reads not able to be characterized by Squeegee (e.g., they could either represent microbial contaminants or bonafide metagenomic signal). In order to detect low-abundance contaminants, using Decontam with negative control samples is recommended.

Squeegee tracks contaminants that came from the same source, such as DNA extraction kits or laboratory surfaces, and in order for Squeegee to perform well, the input samples should be collected from different ecological communities, for example, the microbiota of well-distinct body sites (skin, gut, and oral). One of the other limitations of Squeegee is that it cannot trace contaminants originating from the sample collection process since different sample collection operations

**Table 2 | Tools comparison on handling contamination from a different source**

|  | Squeegee | Recentrifug-e[30] | Decontam[3] | DecontaMine-r[43] | Conterminator[23] |
|---|---|---|---|---|---|
| Lab environment | ✓ | ✓ | ✓ |  |  |
| Lab reagents | ✓ | ✓ | ✓ |  |  |
| Taxonomic classification errors | ✓ |  |  |  |  |
| Cross-contamination | ✓ | ✓ |  |  |  |
| DNA from host/human |  |  |  | ✓ |  |
| Contaminated database |  |  |  |  | ✓ |
| Negative control free | ✓ |  | ✓(*) | ✓ | ✓ |

*Decontam requires either auxiliary DNA quantitation data or negative control data.

may introduce different contaminant species. Therefore, further investigation is required to validate whether the species truly originated from the sampled metagenome for species that are not included in the predicted contaminants.

A stable community member of a specific body site has the potential also to be a contaminant taxon from an external source. Since Squeegee operates without prior knowledge of the input dataset, ubiquitous species that are commonly found in a wide range of environments could allow Squeegee to make false predictions. Although the *Staphylococcus* genus has been reported as external contamination from multiple studies, it is hard to ignore that some of the *Staphylococcus* species may be truly present among multiple body sites, including skin and nasal samples. Such ubiquitous species may introduce noise in Squeegee's predictions. Combined with the prior knowledge of the input dataset and the comprehensive information that Squeegee outputs, the user may further filter the predicted list of contaminants if needed. For any individual sample type, the user should treat the predicted result with care to avoid potential community members being falsely labeled as contaminants. We looked closer into the low weighted precision of non-control sample abundance at species and genus rank for Squeegee in the maternal/infant dataset. *Streptococcus mitis*, a common member of the microbial communities from the oral, skin, female genital tract, and gastrointestinal tract[41,42], was incorrectly identified as a contaminant. Given the high relative abundance of *Streptococcus mitis* (Fig. 3), this false positive contaminant species received an abnormally high weight compared to the other true positive contaminant species, lowering Squeegee's weighted precision. At genus rank, the relative abundance difference between Streptococcus and the true positive putative contaminant genera becomes even greater, which explains why Squeegee genus rank performance is lowered compared to species rank in this dataset. These results highlight areas for future improvement to Squeegee that would allow it to take into account microbial species ubiquitous in many different environments.

By no means is Squeegee meant to be a replacement for experimental negative controls. It does not estimate the relative abundance of each predicted potential contaminant since the relative abundance of the contaminants varies in different sample types. Squeegee makes predictions based on the assumption that the input data are sampled from multiple distinct microbiomes, and does not apply to cases where the sequencing data are from similar microbiomes. If possible, performing negative control experiments will likely provide a more accurate profile of the external contaminants. However, as discussed, it is common for experimental negative control samples to be unavailable for publicly available metagenomic datasets. The metagenomic datasets from the Human Microbiome Project is one such high-profile example. When compared to other contamination removal methods, Squeegee is the only existing tool able to predict contamination from multiple

sources without experimental negative control samples (see Table 2), and its contaminant predictions can have a significant impact on diversity measures which are often a key part of the results of a vast range of microbial studies[3,23,30,43,44].

Another possible solution for contamination detection without negative control sequencing data is to use a contaminant database. If a database of genomes containing known contaminant species exists, we could identify the contaminant sequences in the data by mapping reads against this database[45]. Building such a contaminant database can be challenging because it requires sequencing data from all possible sources of contamination. Since Squeegee is a negative control-free tool for identifying novel contaminants, it can also be used as an important step in filling out such a comprehensive database of likely putative contaminants.

Over 81% of the contaminant species predicted by Squeegee for the HMP dataset match the bacterial genus described as inherent contaminants of the MoBio DNA extraction kit, which was used for the Human Microbiome Project[18]. The cumulative relative abundance of correct prediction is 68.6%, while Squeegee failed to predict most of the genera from phylum *Proteobacteria*. This may be due to the fact that the kit used in the Mobio contamination study[18] is closely related to the one used for HMP but not identical. The contamination profile of the same kit might change over time, and samples processed in different labs may also affect the results since contaminants from lab surfaces and lab members can potentially contribute to the composition of the contamination.

Finally, though Squeegee was tested and evaluated with metagenomic shotgun sequencing datasets, it could be extended for use on 16S rRNA sequencing data. However, Squeegee wouldn't be able to use the breadth and depth of genome coverage of the alignment to determine classification errors. Therefore, choosing an accurate taxonomic classifier is critical for running Squeegee on 16S rRNA sequencing data.

In summary, as far as we are aware, Squeegee is the first de novo computational method for identifying potential microbial contaminants in microbiome datasets in the absence of environmental negative control samples and auxiliary information such as DNA concentration information. Squeegee predictions on multiple datasets have shown that contaminant sequences from the same source, such as DNA extraction kits and other reagents used during the sample processing and sequencing, can be accurately identified across multiple samples using this computational method without experimental negative controls or DNA quantitation data. Squeegee achieves high weighted recall (weighted by both relative abundance of taxa in negative control and non-control samples) and low false positive rates on real metagenomic datasets, and can help to identify putative contaminant sequences of suspicious taxa for low-biomass microbiome studies, enabling sample-independent and orthogonal approaches aimed at distinguishing true microbiome signals from environmental contamination.

## Methods

### Samples from distinct environments

In order to generate reproducible estimates of contaminants and their composition among the samples, the user must collect sequencing data from multiple metagenomic samples. The microbial community composition should be largely distinct between any two samples included in the analyses. Here, distinct refers to different metagenomic environments or sample types in which it is rare to observe a given microbial species present across most samples. Each sample should be provided with a tag or descriptor that distinguishes the different types of samples (e.g., oral, vaginal, fecal, soil, ocean, etc.).

### Taxonomic classification

Squeegee first performs taxonomic classification using Kraken v1.1.1[46,47] with default settings (k = 31). The reference database for Kraken was built with complete bacterial/archaeal/viral genomes from NCBI RefSeq (Release 202). A classification report is generated for each of the samples. Based on the classification, Squeegee chooses a set of candidate contaminant species based on the prevalence of the species across the samples. The prevalence score is weighted by the number of samples of the same type to avoid bias introduced by an unbalanced number of samples between sample types. Higher prevalence rates of a species indicate that the species is shared by more samples across more sample types, and it is more likely to be a contaminant.

### Metagenomic distance estimation

Squeegee also calculates the metagenomic similarity between the samples using Mash v2.2.2, a tool that estimates the Jaccard index using MinHash[47,48]. This is done by first generating a sketch of each sample (Mash sketch -s 100000 -k 21 -m 2) and then calculating the pairwise Mash distance between all pairs of samples (Mash dist). High Mash distances indicate that the metagenomes of the two samples are more distinct (i.e., there are fewer genera and species shared between the samples). Squeegee weights shared species coming from more distinct samples as more likely to be a contaminant.

### Read alignment and error identification

Squeegee then fetches the representative genomes for each of the candidate contaminant species from the NCBI RefSeq database used to build the Kraken database. These representative genomes are used as references to perform a multi-alignment for all reads in the samples using Bowtie2 v2.3.5 with the multi-alignments enabled (bowtie2 –local -a –maxins 600)[47,49]. To accelerate this process, k-mer-mask from meryl v1.0 is used to filter out reads that do not contain any 28-mers from the reference genomes (k-mer-mask -ms 28 -clean 0.0 -match 0.01 -nomasking)[47,50]. Based on the alignment results, the breadth and depth of genome coverage is calculated for each of the sample type using samtools v1.11 (samtools depth)[47,51]. The breadth and depth of genome coverage are used to determine whether the species is truly present or is a potential misclassification from the taxonomic classifier. A species that is truly present should have a large proportion of its genome covered. On the other hand, a large number of reads covering only a small proportion of the genome often suggests that the species was a misclassification[45]. Since contaminant species are often low in abundance, combining samples from the same type would give us a better indication of the presence of the species.

### Contaminant predictions

In the last step, Squeegee combines multiple pieces of evidence, including the prevalence score, Mash distance score, and alignment score, and makes a final prediction for contaminant species using the following equation,

$$C_i = \frac{1}{3} \cdot \left( \frac{P_i}{\overline{P_i}} + \frac{M_i}{\overline{M_i}} + \min\left( \frac{A_i}{5 \cdot \text{min\_cov}}, 1 \right) \right) \qquad (1)$$

In Eq. (1) above, $P_i$ is defined as the prevalence score of candidate contaminant species $i$, which is calculated as the weighted mean prevalence rate of species $i$ among all sample types. $M_i$ is the Mash distance score of candidate contaminant species $i$. Squeegee takes the Mash distance values (from 0 to 1) of all sample pairs that both contain species $i$, and calculates $M_i$ by averaging the top 10% of the pairwise Mash distance value. $A_i$ is the alignment score of candidate contaminant species $i$, which is defined as the mean breadth of genome coverage of species $i$ across sample types with a minimum depth of coverage of 3. $min\_cov$ is the minimum coverage threshold defined by the user.

While calculating the combined contaminant score of each taxon, both the prevalence score and Mash score are normalized by the mean of those scores for all candidate contaminant species ($\overline{P_i}$ and $\overline{M_i}$). The alignment score is capped at 1 for those taxa which have a mean breadth of genome coverage exceeding 5× of the minimum breadth of coverage threshold ($min\_cov$) since the breadth of genome coverage can vary greatly between species. For example, if the minimum breadth of coverage is set to 5%, taxa with a mean breadth of coverage exceeding 25% will receive an alignment score of 1, and taxa with a mean breadth of coverage of 5% will receive an alignment score of 0.2, and taxa with a mean breadth of coverage less than 5% will be classified as false calls and be eliminated by the filter. Details about how each score is calculated can be found in Supplementary Table 3.

Such a scoring mechanism allows Squeegee to automatically distribute different weights based on which evidence contributes more to distinction for the candidate contaminants. For example, if all candidate contaminants have similar Mash scores and similar breadth of genome coverage, but distinguishable prevalence across different sample types, then the algorithms for the contaminant predictor would be automatically favoring prevalence over other factors.

After the combined contaminant scores are calculated, Squeegee filters out species below a user-defined minimum combined score threshold. The combined score averages all three normalized scores for each piece of contaminant evidence. Candidate contaminants with a low combined score suggest insufficient evidence supporting the argument that the candidate species is both an actual contaminant and present in the samples. Squeegee also provides a comprehensive output for the user if the further downstream analysis is required.

The parameter settings retain the potential to affect the precision and recall of Squeegee. Based on the basic understanding of the samples, the user is able to control how likely a taxon is to be recruited as a candidate contaminant by setting a minimum prevalence threshold (Default:0.6) to different values. If the users are processing samples that have similar microbiome communities, increasing the minimum prevalence threshold will reduce the number of false positives caused by shared true community members. Lowering the minimum prevalence threshold allows the program to consider more candidate contaminants, potentially increasing recall but will increase the run time. The minimum read support threshold, minimum abundance threshold, and minimum alignment coverage threshold all contribute to how restrictive a taxon is considered present. Based on different sequencing technologies, more than 5% of the reads may be misclassified by the taxonomic classifier even at the genus level[35,46,52]. Increasing those thresholds allows more confident identification of whether a taxon is truly present or not. On the other hand, in a scenario where contaminant species are low abundant, setting those parameters at high values could cause an increase in false negatives.

## Evaluation of Squeegee

Evaluation of Squeegee predictions was performed by comparing the predicted contaminant species using three datasets: (1) a simulated dataset with ground truth contaminant species, (2) a real dataset with available negative control samples, and (3) a real dataset without a negative control (HMP samples) but with associated kit contaminants. The simulated dataset contains a total of 126 simulated samples representing seven distinct microbial communities. The real dataset contains 344 samples over 9 distinct sample types collected of adult females and infants, as well as sequencing data from ten negative control experiment samples. The HMP dataset includes 749 samples collected and sequenced from healthy individuals across 16 different body sites. The parameters and data characteristics are shown in Supplementary table 1.

For (1), the simulated dataset, the contaminant species in the ground truth were generated based on the species of a simulated spike-in of contaminant sequences. In order to simulate a realistic dataset and test the detection limit of Squeegee, 42 real-world metagenomic samples were chosen from seven distinct environments, including six soil samples of mining sites, one soil sample collected from the wetland, six freshwater samples, seven hot spring samples, six skin samples of cows, ten healthy human skin samples, and six healthy human gut samples.[18,53–57] We filtered out the species with relative abundance lower than 0.0005 or with supported read count less than 300 in those samples, and used the remaining species and their relative abundance as a reference to simulate the dataset. We then used the species with relative abundance greater than 0.01 found in the FastDNA SPIN Kit for Soil (MP Biomedicals) from the previous study[2] to simulate contaminant sequences. Each distinct sample was simulated three times with spiked-in contaminant sequences that occupy 0.25, 0.5, and 1% of the total sequences in the sample. A total of 126 simulated samples were generated using CAMISIM and ART, simulating Illumina paired-end reads with an average read length of 150 bp[58,59] and an average read pair count of 6664348. The simulated samples are grouped into three groups by the spike-in contaminant level, and we evaluate Squeegee on each of the groups individually.

For (2), maternal/infant metagenomic datasets, the contaminant species in the ground truth were generated based on the classification of multiple experimental negative controls. To minimize classification errors, we applied a set of criterion to include a species in the contamination ground truth. Species with relative abundance above 0.5% and more than 20 reads assigned in at least half of the negative control samples and species with relative abundance above 10% in a single sample were chosen for inclusion in the ground truth contaminant set, resulting in a strict ground truth set with 15 species from 13 genera. We also applied more permissive filtering and generated a permissive ground truth contaminant set that contains 31 species that belong to 16 genera by lowering the minimum relative abundance threshold to 0.2%. We then aligned the sequencing reads in the experimental control samples to the representative genomes. Reads assigned to the *Staphylococcus virus Andhra* stacked in a small 449 bp region with an average depth of 1429, indicating a false classification call, so we removed it from both strict and permissive ground truth contaminant sets. Once the ground truth contaminants were identified, the relative abundance of the ground truth contaminants was calculated as the average relative abundance across all negative control samples over the sum of the average relative abundance of each contaminant.

For (3) the HMP dataset, which was extracted using the MoBio DNA extraction kit[60], we used the 61 bacteria genus (excluding lot-dependent organisms), which were identified as inherent contaminants within a latter version of a related MoBio extraction kit, the MoBio PowerMax® Soil DNA Isolation Kit 12,988-10 (MoBio Laboratories, USA), in a recent study[18] as the ground truth contaminants. Relative abundances of each genus were also obtained in the same

study. Since Squeegee makes contamination predictions at the species level, predicted contaminant species from the reference genus are counted as true positives. During the evaluation, *Cutibacterium acnes* (formerly *Proprionibacterium acnes*) was assigned to the genus *Proprionibacterium* to keep the ground truth and Squeegee prediction consistent.

Performance is evaluated via precision, recall, and F-score. The unweighted precision is calculated as the ratio between the number of predicted contaminants found in the ground truth and the total number of predicted contaminants. The unweighted recall is calculated as the ratio between the number of correctly predicted contaminants and the total number of contaminants in the ground truth. The unweighted F-score is calculated as 2 × (unweighted precision × unweighted recall)/(unweighted precision + unweighted recall). Those three measurements are also calculated using the mean fraction of the reads of each taxon in the non-control samples as weight. The weighted precision is calculated as the average fraction of reads from the non-control samples in the true positive (TP) taxa over those in the true positive and false positive (TP + FP) taxa. The weighted recall is calculated as the average fraction of reads from the non-control samples in the true positive (TP) taxa over those in the true positive and false negative (TP + FN) taxa. The weighted F-score is calculated as 2 × (weighted precision × weighted recall)/(weighted precision + weighted recall). We also evaluated the methods using the cumulative relative abundance of true positive taxa, which is a weighted recall score that is weighted by the mean relative abundance of the taxa in the negative control samples.

## Reagent-specific contamination detection in human RNA-seq data

To further demonstrate the contamination detection capabilities of Squeegee, we leveraged a human-derived RNA-seq dataset from a study performed by the Genetic European Variation in Health and Disease (GEUVADIS) consortium. The dataset contains parallel RNA-Seq samples from Epstein-Barr virus (EBV)-positive lymphoblastoid cell lines that are sequenced across seven different sequencing centers with identical library preparation kits. We used samples from six out of seven sequencing centers that used the Illumina sequencing platform (Illumina Genome Analyzer II) and excluded the ones that used AB SOLiD System 3.0, leaving us with a total number of 40 paired-end sequencing runs.

We then mapped each of the sequencing runs with bowtie2 against the human reference genome (Homo sapiens GRCh38.p13) with the parameter *–maxins 600* to remove human reads. We gathered the unmapped reads for each of the samples and used them as input for Squeegee. The sample type for each of the samples is labeled by the sequencing center where it was run. The parameter settings for Squeegee and data characteristics are shown in Supplementary Table 1.

## Contamination detection using Decontam

In order to benchmark the performance of Squeegee, we also ran Decontam v1.10.0[3] on the maternal/infant metagenomic datasets with the negative control samples. After taxonomic classification with Kraken, all species with at least 30 read support or relative abundance greater or equal to 0.0005 were collected to construct the abundance input table for Decontam. Contamination detection was done using *isContaminant* function with the prevalence method from the Decontam R package with the default parameters.

## Alpha diversity analysis of predicted contaminants

We categorized the labeled sample types of the maternal/infant dataset and HMP dataset into combined sample types based on the body site. The combined sample types for the maternal/infant

dataset include placenta, breast milk, oral, stool, and vaginal. The combined sample types for HMP include vaginal, throat, stool, oral, skin, and nasal samples. Samples from the same combined sample types in each dataset were used for alpha diversity analysis. Both Shannon's diversity index and Simpson's diversity index were measured before and after contamination removal. Only reads assigned to the species rank by Kraken were used in calculating Shannon's diversity index and Simpson's diversity index. Since contamination originating from external sources can also be actual community members of the metagenomes, we set a max removal cutoff and only remove species with relative abundance below this cutoff. The significance test was done using a two-sided Mann–Whitney $U$-test for all combined sample types with more than 20 samples.

### Stable community members for human body sites

We used the samples from the HMP dataset and their combined sample types to generate a set of stable community members for different human body sites. Stable community members were defined as genera with more than 1% of their reads assigned from Kraken classification in more than 50% of the samples from the same combined sample types.

### Reporting summary

Further information on research design is available in the Nature Portfolio Reporting Summary linked to this article.

## Data availability

Source data are provided with this paper and have been deposited in the OSF database with accession AP7CD[61]. All sequencing data supporting the findings of this study is publicly available. The simulated datasets generated in this study have been deposited in the Zenodo database with accession 7064705[62], 7062953[63], and 7064599[64]. The maternal/infant metagenomic datasets are available for download via NCBI BioProject PRJNA725597. The HMP samples are downloaded from http://www.hmpdacc.org/HMASM/. The human RNA-Seq datasets are available for download via NCBI BioProject PRJEB2123. The datasets were simulated/sequenced from distinct samples, and no sample was simulated/sequenced repeatedly.

## Code availability

The source code for Squeegee is publicly available at https://gitlab. com/treangenlab/squeegee, and we used version 0.2.0 of Squeegee[65] for the result and analysis presented in this manuscript. The code used for analysis and figure generation used in this study can be found in the OSF database with accession AP7CD[61].

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

## Acknowledgements

We would like to thank Michael Nute for their constructive comments and helpful feedback. We would also like to thank Ben Callahan for his feedback on the Decontam algorithm. Special thanks to the depositors of all publicly available metagenomic datasets used in this study.

T.J.T. was supported in part by NIH grant P01-AI152999 from the the National Institute of Allergy and Infectious Diseases (NIAID) and NSF grant EF 2126387. R.A.L.E., Y.L., and T.J.T. were supported by the FunG-CAT program from the Office of the Director of National Intelligence (ODNI), Intelligence Advanced Research Projects Activity (IARPA), via the Army Research Office (ARO) under Federal Award No. W911NF-17-2-0089. R.A.L.E. was supported in part by a training fellowship from the Gulf Coast Consortia, on the NLM Training Program in Biomedical Informatics & Data Science (T15LM007093). This research was made possible in part by an NIH-funded fellowship to Dr. Michael Jochum (T32 HD098069). This work was also supported in part by the Big-Data Private-Cloud Research Cyberinfrastructure MRI award funded by NSF under grant CNS-1338099 and by Rice University's Center for Research Computing (CRC). Additional support for this work arose from the NIH

(NIH-NINR R01NR014792, NIH-NICHD R01HD091731, NIH-NIDDK 1R01DK128187-01A1, all to K.M.A.).

## Author contributions

R.A.L.E., K.M.A., and T.J.T, conceived the experiment(s), Y.L. conducted the experiment(s), M.D.J., R.A.L.E., K.M.A., T.J.T., and Y.L. analysed the results. All authors reviewed the manuscript.

## Competing interests

The authors declare no competing interests.
