## [Peer Review File · Nature Communications]

Squeegee: Identifying Contaminants in Low Microbial Biomass Microbiomes when Negative Controls are UnavailableReviewers' comments:

Reviewer #1 (Expertise: statistical methods for the removal of microbial contaminants):

“Squeegie: de-novo identification of reagent and laboratory induced microbial contaminants in low biomass microbiomes” describes a new computational tool -- Squeegie -- that identifies contaminants in shotgun metagenomic data from low biomass environments. The authors validate Squeegie in a simulated dataset with a known ground truth, and in two real datasets in which the ground truth contaminants are determined by negative controls (held out from Squeegie testing) or by previous investigation of the contaminant taxa in the relevant sequencing kits. The authors report modest-to-good positive identification of contaminating taxa (30-80%) and somewhat higher identification when weighting contaminants by abundance (50-90%). Squeegie provides a new functionality within the world of microbiome analysis by allowing contaminant ID in shotgun metagenomic datasets without negative controls. This is particularly relevant to previously generated studies available from public repositories that often lack negative control sequencing information.

I was not convinced that there is a large use case for Squeegie justified here. The manuscript eschews any head-to-head benchmarking with other contaminant ID methods, focusing instead on Squeegie's somewhat unique functionality to make contaminant IDs in the absence of negative controls. But how broad is this niche? As the authors state explicitly, and acknowledge implicitly in their use of negative controls as a ground truth, negative controls are a preferred means to identify contaminants in low biomass samples. Going forward, it would be a mistake for low biomass studies to fail to include them. So is the value of Squeegie basically restricted to retrospective re-analysis of old studies? And only old shotgun studies at that?

Some of the ideas behind Squeegie are interesting, for example using MASH distances between samples to weight the evidence for a contaminant origin of taxa shared between them. But, the details of the method are not sufficiently described. In that example, how is such evidence weighted? But that problem is more widespread, none of the parameters going into Eq 1 are appropriately described. We should know how every variable in the critical equation describing the classifier is calculated, and as far as I can tell the manuscript doesn't describe in detail how any of them were calculated.

In the evaluation of Squeegie, Precision, Recall and Weighted Recall were all provided. Why was there no Weighted Precision provided? One of the big concerns about contaminant identification is False Positives, and the way Squeegie works the concern is that the False Positives might be real taxa that occur at high relative abundance across many samples -- i.e. major parts of the study-wide microbiome. For example, *Strep mitis* in Fig 3, and as noted by the authors in the Discussion. The concern here is increased by the additional cutoff the authors used when analyzing the HMP data -- they chose to ignore contaminant IDs from Squeegie that were at high relative abundances. Because there were abundant false positives?

Describing a way for users to evaluate and choose a contaminant classification threshold would go a long way towards making this a more useful tool for other users. Also how to evaluate whether multiple microbiomes are sufficiently “distinct” so that Squeegie is effective.

Reviewer #2 (Expertise: computational methods for the removal of microbial contaminants in databases):

Liu et al. propose Squeegee, a method to filter contamination from metagenomic samples without requiring a negative control. It relies on diverse samples as input to find consensus species occurring across all samples, which have a higher chance to be lab contaminants.

The method reports a contaminant score per species that combines multiple sources of evidence. The software is open-source and packaged in Bioconda. Squeegee was evaluated using simulated data, real data with negative controls and real data without negative controls. The method has merits and tackles an important problem. My main concerns are related to the benchmarks.

Major:

(1) The study "Human placenta has no microbiome but can contain potential pathogens" by de Goffau et al. in Nature showed that the placenta should not contain microbes. The authors benchmarked with placenta samples and showed that the alpha-diversity decreases after running Squeegee on them. How many microbes in the placenta samples are classified as contaminants pass the method? I would expect that to achieve a high precision nearly all microbes would have to be classified as contaminants.

(2) The simulated benchmark consists of three samples that contain 5 different bacteria as actually present organisms and 10 common contaminant bacteria genomes. How are the bacteria distributed among the three samples? Are all contaminants in all samples? If yes, then this seems to be overly simplified, as it seems rather easy to identify this common subset if all members are always present.

(3) When benchmarking with negative controls the authors propose abundance thresholds to remove contaminant species from the negative control to reduce the classification error. Why does the abundance matter here? For example, when dealing with pathogens even weak signals (10-100 reads) must often be considered. The full set of negative control is very useful even if the signal is weak. Please also provide a benchmark without rejecting species from the negative control

(4) In the section "Samples from distinct environments" it is mentioned that the method requires diverse samples as input. Do these samples have to be from the same laboratory using the same protocol? How would Squeegee perform on samples where experimental methods like DNA extraction, etc. differ a lot? I assume it might be hard for the method to detect contamination, if the laboratory conditions vary a lot across individual samples. The authors benchmarked on data from the HMP project, which was generated by different laboratories. However the HMP samples follow very stringent protocols, leading to very little technical variance. I would like to see the performance of Squeegee on less consistent samples.

Minor:

(1) In Figure 5 *Cutibacterium acnes* produces the highest contamination score while being a false positive. The alignment score seems very high. Based on the distribution I would assume it should be much lower. What is the reason for the high scoring FP?

(2) Squeegee combines multiple features to a single score to determine if some species is a contaminant or a real signal.

How much does each of these terms contribute to the prediction performance?

What is the range of the score? How does the score relate to precision or sensitivity?

Providing a fitting from the score to sensitivity/precision would be a great help for users to set this complex parameter.

(3) How does contamination in the DNA database affect Squeegee? Many pathogen genomes contain human DNA contaminants. If performing metagenomic pathogen detection in humans then

many pathogen genomes will occur in samples. However, this might be due to human DNA in their genomes. Would these pathogens be labeled as contaminants?

(4) Typo in Figure 1 "Contamiant_score" should be "Contaminant_score"

(5) Please mention the conda package in the manuscript and cite bioconda.

(6) Metagenomic samples can be large and run time can be critical for users. Please discuss the resource and runtime requirements of Squeegie.

(7) Please open source all scripts and data to make the study reproducible

Reviewer #3 (Expertise: computational methods for single-cell data analysis, multi-modal data integration):

The manuscript entitled "Squeegie: de-novo identification of reagent and laboratory induced microbial contaminants in low biomass microbiomes" gives a free computational method "Squeegie", which can identify potential contaminants, and the manuscript focuses on testing it. But in this article, some problems remain.

1 This study mainly detects microbial contaminants through "Squeegie", but the data is from Human Microbiome Project, we expect authors to provide their original metagenomic data to test "Squeegie".

2 For the HMP dataset at genus rank, "Squeegie" achieved a precision of 0.5 and a recall of 0.328. We hope that authors can improve "Squeegie" to achieve higher precision and recall.

3 Figure 6, Supplementary Fig. 2, and Supplementary Fig. 3 show different results when the max removal cutoff is set to 1% and 0.5%, please explain it. And why is the max removal cutoff set to 0.5% ?

LEGEND

Responses to reviewers in **bold black**
Changes to the manuscript in **blue**
Location of inserted text in (parenthesis)
Added/updated display items indicated in **red**
Reviewers' comments in black

We thank the editor for providing the opportunity to submit a revised version of our manuscript. We have carefully addressed the reviewers' concerns which have strengthened our study and better highlight the performance of our method.

REVIEWER COMMENTS

Reviewer #1 (Remarks to the Author)

"Squeegee: de-novo identification of reagent and laboratory induced microbial contaminants in low biomass microbiomes" describes a new computational tool -- Squeegee -- that identifies contaminants in shotgun metagenomic data from low biomass environments. The authors validate Squeegee in a simulated dataset with a known ground truth, and in two real datasets in which the ground truth contaminants are determined by negative controls (held out from Squeegee testing) or by previous investigation of the contaminant taxa in the relevant sequencing kits. The authors report modest-to-good positive identification of contaminating taxa (30-80%) and somewhat higher identification when weighting contaminants by abundance (50-90%). Squeegee provides a new functionality within the world of microbiome analysis by allowing contaminant ID in shotgun metagenomic datasets without negative controls. This is particularly relevant to previously generated studies available from public repositories that often lack negative control sequencing information.

R1Q1. I was not convinced that there is a large use case for Squeegee justified here. The manuscript eschews any head-to-head benchmarking with other contaminant ID methods, focusing instead on Squeegee's somewhat unique functionality to make contaminant IDs in the absence of negative controls.

We thank the reviewer for the feedback. In order to address this concern, we have added a benchmark experiment by comparing Squeegee with one of the currently used state-of-the-science putative contamination detection tools, Decontam, on the maternal/infant dataset.

> (Result, Benchmark against Decontam, Page 6)

We also benchmarked Squeegee with Decontam, one of the most widely used contamination detection tools. Figure 2 shows the recall and weighted recall of both tools at genus rank and species rank. Decontam achieved a recall of 0.714 (10/14) at genus rank, and a recall of 0.562 (10/16) at species rank. Both Squeegee and Decontam are able to identify the majority of the contaminants at genus rank with weighted recall of 0.937 and 0.916. At the species rank, Squeegee outperformed Decontam with a higher weighted recall of 0.833, compared to Decontam's weighted recall of 0.613. Since the ground truth contaminants used in this comparative evaluation are constructed with criteria that was largely limited to high confidence contaminants, Decontam archives much lower precision than Squeegee.

> (Discussion, Page 9)

During the benchmark comparing Squeegie and Decontam, we tested Decontam with its prevalence-based method, which requires negative control samples as input. Squeegie still outperformed Decontam in both recall and weighted recall at genus and species ranks. It is possible that Decontam did not flag neither those genera nor species as contaminants that are shared between source of contamination and the sampling environment.

> (Methods, Contamination Detection using Decontam, Page 12)

In order to benchmark the performance of Squeegie, we also ran Decontam v1.10.0 on the maternal/infant metagenomic datasets with the negative control samples. After taxonomic classification with Kraken, all species with at least 30 read support or relative abundance greater or equal to 0.0005 were collected to construct the abundance input table for Decontam. Contamination detection was done using *isContaminant* function with the prevalence method from the Decontam R package with the default parameters.

Figure 2. Squeegie and Decontam accuracy at genus and species ranks. The precision is calculated as the ratio between the number of predicted contaminant taxa found in the ground truth and the total number of predicted contaminant taxa. The recall is calculated as the ratio between the number of predicted contaminant taxa found in the ground truth and the total number of taxa in the ground truth. The weighted recall is calculated as the proportion of the reads assigned to the correctly predicted taxa over the total number of reads assigned to the ground truth contaminant taxa.

R1Q2. But how broad is this niche? As the authors state explicitly, and acknowledge implicitly in their use of negative controls as a ground truth, negative controls are a preferred means to identify contaminants in low biomass samples. Going forward, it would be a mistake for low biomass studies to fail to include them. So is the value of Squeegie basically restricted to retrospective re-analysis of old studies? And only old shotgun studies at that?

We appreciate the reviewer's concern related to the breadth of application of Squeegie. We also agree that our intent is not to encourage forgoing negative controls. First, we have changed the title to highlight the exact use case we envision Squeegie being most useful: **"Identifying Contaminants in Low Microbial Biomass Microbiomes when Negative Controls are Unavailable"**.

Next, to highlight that Squeegie is not limited to detecting microbial contaminants in metagenomic shotgun sequencing studies, we have added a new set of experiments in which Squeegie is applied to human-derived single cell RNA-seq data.

>(Methods, Reagent Specific Contamination Detection in Human RNA-Seq Data, Page 12)

To further demonstrate contamination detection capabilities of Squeegie, we leveraged a human-derived RNA-seq dataset from a study performed by the Genetic European Variation in Health and Disease (GEUVADIS) consortium. The dataset contains parallel RNA-Seq samples from Epstein-Barr virus (EBV)-positive lymphoblastoid cell lines that are sequenced across 7 different sequencing centers with the identical library preparation kits. We used samples from 6 out of 7 sequencing centers that used Illumina sequencing platform (Illumina Genome Analyzer II), and excluded the ones that used AB SOLiD System 3.0, leaving us with a total number of 40 paired-end sequencing runs.

We then mapped each of the sequencing runs with bowtie2 against the human reference genome (*Homo sapiens* GRCh38.p13) with the parameter `--maxins 600` to remove human reads. We gathered the unmapped reads for each of the samples and used them as input for Squeegie. The sample type for each of the samples is labeled by the sequencing center where it was run. The parameter settings for Squeegie and data characteristics are shown in Supplementary table 1.

> (Results, Reagent Specific Contamination Detection, Page 6)

We applied Squeegie to a human-derived RNA-Seq dataset from an index study that aimed to evaluate the potential for contamination arising during sequencing across 6 different sequencing centers in Europe in the GEUVADIS consortium. The resultant generated RNA sequencing data arose from 40 sequencing runs performed at all 6 sequencing centers on identical sequencing platforms following extraction with the same kits on parallel samples. The prediction from Squeegie indicated that 7 species (*Human gammaherpesvirus 4*, *Proteus virus Isfahan*, *Escherichia coli*, *Bacillus megaterium*, *Bacillus cereus*, *Klebsiella pneumoniae*, *Cutibacterium acnes*) are reagent specific contaminants and can be found in the sequencing runs across different sequencing centers (Supplementary Fig. 6). While *Human gammaherpesvirus 4* is associated with the cell line used during the sequencing, *Escherichia coli*, *Bacillus megaterium*, *Bacillus cereus*, *Klebsiella pneumoniae*, *Cutibacterium acnes* can be putative common “kit contaminants” that have been previously reported. However, *E. coli* and *K. pneumoniae* are also prevalent environmental and human commensal microbes or pathobionts.

> (Discussion, Page 7)

In addition to identifying microbial contaminants within microbiome datasets that lack negative controls, Squeegie can also be used to identify contaminants in human RNA-seq data. In our experiments with the GEUVADIS consortium human RNA-Seq dataset, Squeegie predicted that seven species were lab preparation related. In this dataset, since non-human reads are all classified as contamination, by identifying reagent specific contaminants that are shared across different sequencing labs, one can also backtrack and identify lab specific contaminants using the classification report provided by Squeegie.

Supplementary Figure 5. Predicted potential contaminant species for the human RNA-Seq dataset.

This figure shows the prevalence, the breadth of genome coverage, and additional scoring information of predicted contaminant species. The first 6 columns show the prevalence of each species among each of the sequencing labs, where zero prevalence is marked in blue. The next 6 columns show the breadth of genome coverage of each species in each of the sequencing labs, where zero is marked in blue. The remaining rows show the prevalence score, the alignment score, the Mash score, and the combined score used to make the final prediction.

We have also added the following text suggesting that Squeegie could detect batch specific contamination as well as cross contamination. In both cases, negative control samples might not fully address the problem.

> (Discussion, Page 8)

Another use case for Squeegie is to detect batch specific contaminants, as well as cross contaminants. If negative control samples are not prepared and sequenced for every batch run, it is possible that batch specific contaminants get mixed in the samples, causing bias in the downstream analysis. A similar scenario is that cross contamination occurs in a single batch but happens to not affect the negative control sample. Running Squeegie on each individual batch allows the user to detect such batch specific contaminants or cross contaminants, since Squeegie detection method does not depend on the negative control profiles.

R1Q3. Some of the ideas behind Squeegie are interesting, for example using MASH distances between samples to weight the evidence for a contaminant origin of taxa shared between them. But, the details of the method are not sufficiently described. In that example, how is such evidence weighted? But that problem is more widespread, none of the parameters going into Eq 1 are appropriately described. We should know how every variable in the critical equation describing the classifier is calculated, and as far as I can tell the manuscript doesn't describe in detail how any of them were calculated.

We thank the reviewer for bringing this important point to our attention. The algorithm for combining different evidence to enable a final prediction for contaminant species in the contaminant predictor uses a dynamic weighting strategy, where the equation is constructed by normalizing the prevalence score and mash score using the mean across all candidate contaminants, and by capping alignment score based on parameter setting for the minimum breadth of genome coverage. This allows Squeegiee to make the final prediction favoring the more distinguishable evidence among the candidates in an explainable, reproducible, and transparent way.

We updated/added the following text to provide a more detailed description of how scoring works in the manuscript.

$$C_i = \frac{1}{3} \cdot \left(\frac{P_i}{\bar{P}_i} + \frac{M_i}{\bar{M}_i} + \min \left(\frac{A_i}{5 \cdot \text{min_cov}}, 1 \right) \right)$$

> (Method, Contaminant predictions, Page 10-11)

In the equation above, P_i is defined as the prevalence score of candidate contaminant species i , which is calculated as the weighted mean prevalence rate of species i among all sample types. M_i is the Mash distance score of candidate contaminant species i . Squeegiee takes the Mash distance values (from 0 to 1) of all sample pairs that both contain species i , and calculates M_i by averaging the top 10% of the pairwise Mash distance value. A_i is the alignment score of candidate contaminant species i , which is defined as the mean breadth of genome coverage of species i across sample types with minimum depth of coverage of 3. min_cov is the minimum coverage threshold defined by the user.

While calculating the combined contaminant score of each taxon, both prevalence score and mash score is normalized by the mean of those scores for all candidate contaminants (\bar{P}_i and \bar{M}_i). The alignment score is capped at 1 for those taxons which have mean breadth of genome coverage exceeding 5x of the minimum breadth of coverage threshold (min_cov), since breadth of genome coverage can vary greatly between species. For example, if the minimum breadth of coverage is set to 5%, taxa with mean breadth of coverage exceeding 25% will received an alignment score of 1, and taxa with mean breadth of coverage of 5% will received an alignment score of 0.2, and taxa with mean breadth of coverage less than 5% will be classified as false calls and be eliminated by the filter. Details about how each score is calculated can be found in Supplementary Table 3.

Such scoring mechanism allows Squeegiee to automatically distribute different weight based on which evidence contributes more to distinguishable for the candidate contaminants. For example, if all candidate contaminants have similar mash scores and similar breadth of genome coverage, but distinguishable prevalence across different sample types, then the algorithm for the contaminant predictor would be automatically favoring prevalence over other factors.

> (Method, Contaminant predictions, Page 11)

After the combined contaminant scores are calculated, Squeegiee filters out species that are below a user defined minimum combined score threshold. The combined score averages all 3 normalized scores for each piece of contaminant evidence.

We added the following supplementary table to show how each score is calculated in detail.

Supplementary Table 3. Contaminant Score Calculation

Calculation	
Prevalence Score	For each sample type, calculate the prevalence rate $p_{i,t}$ of species i in sample type t as: $p_{i,t} = \frac{\text{occurrence of } i \text{ in sample type } t}{\text{sample count of } t}$ Then calculate mean prevalence for i as: $P_i = \frac{\sum_{t=1}^T p_{i,t}}{T}, \text{ where } T \text{ is the total number of sample types.}$
Mash Score	For each sample type, calculate the paired-wise mash distance m_i of species i for all samples that species exist. Then calculate mash score for i as: $M_i = \text{mean of } m,$ where m is a subset of m_i such that element in m is above 90th percentile of all m_i
Alignment Score	For each sample type, calculate the breadth of genome coverage $a_{i,t}$ of species i in sample type t as: $a_{i,t} = \frac{\text{length of } G_i \text{ with at least 3 reads covered in sample type } t}{\text{total length of } G_i},$ where G_i is the reference genome of species i. Then calculate alignment score for i as: $A_i = \frac{\sum_{t=1}^T a_{i,t}}{T}, \text{ where } T \text{ is the total number of sample types.}$
Combined Score	The combined score is calculated by normalizing each of the score using the equation below: $C_i = \frac{1}{3} \cdot \left(\frac{P_i}{\bar{P}_i} + \frac{M_i}{\bar{M}_i} + \min\left(\frac{A_i}{5 \cdot \text{min_cov}}, 1\right) \right)$ where \bar{P}_i is the mean of prevalence score of all candidate species, and \bar{M}_i is the mean of mash score of all candidate species, and min_cov is the minimum coverage threshold defined by the user.

We updated the following workflow figure as well.

Figure 1. Squeegie pipeline workflow. Squeegie starts with taxonomic classification using Kraken to determine a set of candidate contaminant species. Reads from the input data are aligned to the representative genomes of the candidate contaminant species using Bowtie2 in multi-alignment mode. It also calculates the pair-wise Mash distance for all the samples. It combines the prevalence, the Mash distance, and breadth/depth of genome coverage of the candidates to predict potential contaminants.

R1Q4. In the evaluation of Squeegie, Precision, Recall and Weighted Recall were all provided. Why was there no Weighted Precision provided?

The recall is calculated as the number of correctly predicted taxa over the number of total taxa in the ground truth set, $TP/(TP+FN)$, where TP is a correctly predicted taxa and FN is a missed taxa. The weighted recall is calculated using the relative abundance of each taxon in the ground truth set as weight, and it represents the sum of relative abundance of the correctly predicted taxa in

the ground truth set. However, since precision calculation is based on the predicted contaminants output by Squeegie, and precision is defined as $TP/(TP+FP)$, where FP is a falsely predicted contaminant, there is no truth relative abundance associated with the FP predictions, so we are unable to calculate weighted precision.

R1Q5. One of the big concerns about contaminant identification is False Positives, and the way Squeegie works the concern is that the False Positives might be real taxa that occur at high relative abundance across many samples -- i.e. major parts of the study-wide microbiome. For example, *Strep mitis* in Fig 3, and as noted by the authors in the Discussion. The concern here is increased by the additional cutoff the authors used when analyzing the HMP data -- they chose to ignore contaminant IDs from Squeegie that were at high relative abundances. Because there were abundant false positives?

The reviewer brings up an important point. Squeegie is designed for identifying potential contaminant species that originated from the same source, such as lab environment and lab preparation kits. The reason behind adding a cutoff while performing alpha diversity analysis for HMP samples is related to ubiquitous species, which are species that can be found in a wide range of environments. Those ubiquitous species can be at the same time contaminants from the external source, and true members of the microbiome community that we sampled from. One of the examples is *Staphylococcus spp.*. *Staphylococcus spp.* has been commonly reported as kit contaminants in multiple previous studies, but it's hard to ignore the fact that it's also a common member of the human microbiome. Without the prior knowledge of the samples themselves, we felt including such ubiquitous species may bias the downstream analysis such as alpha diversity calculation. But in response to the reviewer's concern, we have recalculated the alpha diversity plots without removing the high relative abundance taxa (max removal set to 100%).

Alpha diversity index for maternal/infant dataset. Both Shannon's and Simpson's diversity index of the communities in each of the samples were evaluated before the contaminant reads were removed (red), after removing species only confirmed by the experimental negative control (blue), and after removing all species predicted by Squeegie (black). Numbers inside parentheses are the numbers of samples in each sample type. Significance labeling: n.s.($P > 0.05$), *($P \leq 0.05$), **($P \leq 0.01$), ***($P \leq 0.001$).

Alpha diversity index for HMP dataset. Both Shannon's and Simpson's diversity index of the communities in each of the samples were evaluated before the contaminant reads were removed (red), after removing species only confirmed by the experimental negative control (blue), and after removing all species predicted by Squeegie (black). Numbers inside parentheses are the numbers of samples in each sample type. Significance labeling: n.s.($P > 0.05$), *($P \leq 0.05$), **($P \leq 0.01$), ***($P \leq 0.001$).

For both of the dataset, we observed alpha diversity of certain types of samples changed significantly without maximum removal threshold. For example, in the HMP dataset, both Shannon's and Simpson's index for skin significantly increased, which is as expected, since multiple *Staphylococcus sp* are removed. Another similar scenario resides among stool samples in both of the datasets. Since stool microbiota are microbial high abundance, it is not surprising that it is potentially shared by a large number of members with the environmental/kit

contaminants, removing contaminant taxa with maximum removal threshold is associated with a subsequent observed bias for alpha diversity analysis.

R1Q6. Describing a way for users to evaluate and choose a contaminant classification threshold would go a long way towards making this a more useful tool for other users. Also how to evaluate whether multiple microbiomes are sufficiently “distinct” so that Squeegie is effective.

In addition to the added detailed description of how each of the different pieces of evidence contribute to the contamination detection process (in response to Q1R3), we also added the following paragraph as a guidance for parameter setting in different scenarios and have added additional guidance in the squeegie readme file.

> (Results, Contaminant predictions, Page 11)

The parameter settings retain the potential to affect the precision and recall of Squeegie. Based on the basic understanding of the samples, the user is able to control how likely a taxon is being recruited as a candidate contaminant by setting minimum prevalence threshold (Default:0.6) to different values. If the users are processing samples that have similar microbial communities, increasing the minimum prevalence threshold will reduce the number of false positives caused by shared true community members. Lowering the minimum prevalence threshold allows the program to consider more candidate contaminants, potentially increase recall but will increase the run time. Minimum read support threshold, minimum abundance threshold, and minimum alignment coverage threshold all contribute to how restrict a taxon is considered present. Based on different sequencing technologies, 5% or more of the reads may be misclassified by the taxonomic classifier even at genus level. Increasing those thresholds allows more confident identification of whether a taxon is truly present or not. On the other hand, in a scenario where contaminant species are low in abundance, setting those parameters at high values could cause an increase in false negatives.

Reviewer #2 (Remarks to the Author)

Liu et al. propose Squeegie, a method to filter contamination from metagenomic samples without requiring a negative control. It relies on diverse samples as input to find consensus species occurring across all samples, which have a higher chance to be lab contaminants.

The method reports a contaminant score per species that combines multiple sources of evidence. The software is open-source and packaged in Bioconda. Squeegie was evaluated using simulated data, real data with negative controls and real data without negative controls. The method has merits and tackles an important problem. My main concerns are related to the benchmarks.

Major:

R2Q1. The study "Human placenta has no microbiome but can contain potential pathogens" by de Goffau et al. in Nature showed that the placenta should not contain microbes. The authors benchmarked with placenta samples and showed that the alpha-diversity decreases after running Squeegie on them. How many microbes in the placenta samples are classified as contaminants pass the method? I would expect that to achieve a high precision nearly all microbes would have to be classified as contaminants.

We thank the reviewer for the feedback. Because the reviewer brings up the manuscript of de Goffau et al, we note that this work served as an inspiration in developing our novel tool,

Squeezegee. Namely, despite suggestions in the manuscript to the contrary, de Goffau et al did not actually include negative or kit contaminant controls for their WGS analyses; they were only included in their 16S rRNA sequencing. Because there are numerous robust studies demonstrating to the contrary (namely that the placenta does harbor a low biomass, low diversity metagenome), our team of authors was interested in analyzing the data of de Goffau et al. Upon receipt of the data to do so, it was then evident that customary decontaminant pipelines had not and could not be run on de Goffau's metagenomic dataset because it failed to include kit nor environmental contaminant controls. Parenthetically, it also does not contain other body sites, including vaginal, samples nor data, for comparison. From these limitations was born Squeezegee, and hence de Goffau et al is a great case example for why this tool is important.

As noted in our current manuscript, in order to overcome limitations such as not having kit or environmental contaminant controls, there does have to be common source distinctions. Squeezegee incorporates the same source microbial potential contaminant source. Ergo, one of the limitations of Squeezegee is that it only tracks microbial contaminants from the same source, for example lab environment and reagents used during the sample collection process. Therefore, it cannot trace the contaminants introduced by different sample collection processes. It is possible that true contaminants introduced during sample collection from different sources will not be detected by squeezegee as potential contaminants. This is also mentioned in the text that in some cases, further investigation is required to validate whether the species truly originated from the sampled metagenome even if the species are not included in the predicted contaminants. Furthermore, Figure 3a shows contaminants detected by negative controls that were not detected by Squeezegee (6 total, with 3 under 1% relative abundance):

(a) Maternal/infant dataset.

R2Q2. The simulated benchmark consists of three samples that contain 5 different bacteria as actually present organisms and 10 common contaminant bacteria genomes. How are the bacteria distributed among the three samples? Are all contaminants in all samples? If yes, then this seems to be overly simplified, as it seems rather easy to identify this common subset if all members are always present.

In order to resolve this important issue raised by the reviewer, we replaced the old simulated data experiment in the manuscript with a newly designed simulated dataset. The new simulated dataset is constructed using the relative abundance profiles of real life metagenomic samples, and it is far more complex than the old one and it reflects the characteristics of the microbiome communities in reality. The spike-in contaminant sequence is also simulated based on the relative abundance profile of a real life DNA extraction kit. Details describing this new simulated data experiment have been added to the text.

The following table shows the details of the simulated dataset.

Collection Site	Sample Type	Accession ID of reference sample for simulating relative abundance	Relative abundance of spiked in contaminants	# of simulated samples	Average Read Count	Read Length
Mining Site	Soil Metagenomic	SRR7521490	0.05%, 0.1%, 0.25%, 0.5%, 1%	5	3329844	Paired-end 150bp
Wetland	Soil Metagenomic	SRR8879132	0.05%, 0.1%, 0.25%, 0.5%, 1%	5	3329844	Paired-end 150bp
Freshwater (lakes and ponds)	Water Metagenomic	ERR4413942	0.05%, 0.1%, 0.25%, 0.5%, 1%	5	3329844	Paired-end 150bp
Hot Spring	Water Metagenomic	ERR791764	0.05%, 0.1%, 0.25%, 0.5%, 1%	5	3329844	Paired-end 150bp
Skin	Human Metagenomic	SRR3644400	0.05%, 0.1%, 0.25%, 0.5%, 1%	5	3329844	Paired-end 150bp
Stool	Human Metagenomic	ERR3450203	0.05%, 0.1%, 0.25%, 0.5%, 1%	5	3329844	Paired-end 150bp

> (Methods, Evaluation of Squeegie, Page 11)

For (1), the simulated dataset, the contaminant species in the ground truth were generated based on the species of a simulated spike-in of contaminant sequences. In order to simulate a realistic dataset, six real world metagenomic samples were chosen from distinct environments, including a soil sample of a mining

site, a soil sample collected from wetland, a fresh water sample, a hot spring sample, a healthy human skin sample, and a healthy human gut sample. We filtered out the species with relative abundance lower than 0.0001 or with read support less than 300 in those samples, and used the remaining species and their relative abundance as a reference to simulate the dataset. We then used the species with relative abundance greater than 0.0001 found in the FastDNA SPIN Kit for Soil (MP Biomedicals) from the previous study to simulate contaminant sequences. Each distinct environment was simulated 5 times with spiked-in contaminant sequences that occupy 0.05%, 0.1%, 0.25%, 0.5%, 1% of the total sequences in the sample. A total of 30 simulated samples were generated using CAMISIM and ART simulating Illumina paired-end reads with average read length of 150bp and average read count of 3329844.

> (Results, Genus level accuracy of Squeegie prediction, Page 3)

For the simulated dataset, Squeegie achieves a precision of 0.714 (5/7) and a recall of 0.556 (5/9) at genus rank. The weighted recall for the simulated dataset is 0.788, indicating that the genera that Squeegie failed to identify are low abundance among the spike-in contaminant sequences.

> (Results, Species level accuracy of Squeegie prediction, Page 5-6)

Squeegie achieved a precision of 0.727 (8/11) and is able to correctly predict 8 out of 12 potential contaminant species for the simulated dataset at species rank with a weighted recall of 0.788. Detailed information of the performance of Squeegie on the simulation data at both genus and species ranks can be found in Fig. 5.

Figure 5: Squeegie prediction accuracy at genus and species ranks on the simulated dataset. The precision is calculated as the ratio between the number of predicted contaminant taxa found in the ground truth and the total number of predicted contaminant taxa. The recall is calculated as the ratio between the number of predicted contaminant taxa found in the ground truth and the total number of taxa in the ground truth. The weighted recall is calculated as the proportion of the reads assigned to the correctly predicted taxa over the total number of reads assigned to the ground truth contaminant taxa.

R2Q3. When benchmarking with negative controls the authors propose abundance thresholds to remove contaminant species from the negative control to reduce the classification error. Why does the abundance matter here? For example, when dealing with pathogens even weak signals

(10-100 reads) must often be considered. The full set of negative control is very useful even if the signal is weak. Please also provide a benchmark without rejecting species from the negative control.

The main reason for using a combination of minimum abundance while performing taxonomic classification on the samples (both negative control and microbial samples) is to reduce the classification error from taxonomic classification tools. K-mer based taxonomic classifiers often suffer from false positive assignment. Based on different short read sequencing technologies, the false positive rate can be up to 5%, even at genus level (<https://genomebiology.biomedcentral.com/articles/10.1186/gb-2014-15-3-r46>). In practice, kraken classification results usually contain a long tail of falsely assigned taxons with ultra low read count/relative abundance. In addition, with the growth of the RefSeq database, taxonomic classification problems become more and more complex, further reducing the accuracy of the classifiers at each rank (<https://genomebiology.biomedcentral.com/articles/10.1186/s13059-018-1554-6>).

The negative control samples used in the material/infant dataset contain an average of less than 0.5 million read count. Considering the average size of a bacterial genome is 5 million bp, and the sequencing depth for our negative control samples, we are not able to confidently call the existence of a bacteria species for those below the abundance of 0.5%. On the other hand, using a minimum threshold of 0.5% is well within the classifier false positive error rate. Indeed, classification of a microbial genome accurately from 10-100 reads remains an open question, and it is also the reason that we need to use a combined approach including determining prevalence with minimum abundance threshold and breadth and depth of genome coverage calculation with the alignment step in Squeegiee.

R2Q4. In the section "Samples from distinct environments" it is mentioned that the method requires diverse samples as input. Do these samples have to be from the same laboratory using the same protocol? How would Squeegiee perform on samples where experimental methods like DNA extraction, etc. differ a lot? I assume it might be hard for the method to detect contamination, if the laboratory conditions vary a lot across individual samples. The authors benchmarked on data from the HMP project, which was generated by different laboratories. However the HMP samples follow very stringent protocols, leading to very little technical variance. I would like to see the performance of Squeegiee on less consistent samples.

Diversity of the input samples implies different ecological communities, such as microbiota of well-distinct body sites (skin, gut, oral, etc.). The reason for choosing the HMP dataset is that the Human Microbiome Project is a well-controlled study with strict protocols, which allows the method to detect kit specific contaminants with high accuracy. If the HMP datasets were generated with variable protocols, we could still detect laboratory specific contaminants by using samples from the same lab as input, but kit contaminants would indeed be less likely to be detected. Of note, in HMP samples were, in fact, extracted in two separate institutional labs but sequenced at one of four sequencing centers (<https://www.ncbi.nlm.nih.gov/pmc/articles/PMC3574278/>).

We included an additional less consistent human RNA-Seq dataset to track reagent specific contamination. In this dataset, the same samples were sent to six different sequencing centers across Europe for sequencing under well-defined protocols, but the bacteria reads count in the samples varies a lot across different labs

<https://journals.plos.org/plospathogens/article?id=10.1371/journal.ppat.1004437>). The detailed information about this experiment, please refer to R1Q1.

We'd like to highlight that Squeegie was not at all designed as a replacement of the negative control experiments, but it serves as the last line of defense for detecting potential microbial contaminants in absence of negative controls (for example, HMP datasets or the de Goffau et al study), and also can identify contaminants negative controls cannot (batch specific contamination, etc).

We have updated the manuscript with the following text to clarify the input requirement for Squeegie.

> (Discussion, Page 8)

Squeegie tracks contaminants that came from the same source, such as DNA extraction kits or laboratory surfaces, and in order to maximize Squeegie's potential, the input samples should be collected from different ecological communities, for example, microbiota of distinct body sites (skin, gut, and oral).

Minor:

R2Q5. In Figure 5. *Cutibacterium acnes* produce the highest contamination score while being a false positive. The alignment score seems very high. Based on the distribution I would assume it should be much lower. What is the reason for the high scoring FP?

We appreciate the reviewer checking our results. We investigated the issue and found that this was due to the species reclassification in the taxonomy. *Cutibacterium acnes* (formerly *Propionibacterium acnes*) was identified as contaminants by Squeegie with the highest putative contamination score. In the reference article that we used as ground truth, It was also being identified as a major kit contamination but under genus *Propionibacterium*.

We have re-analyzed the HMP dataset with a new set of parameters, and resolved the issue and corrected the results in the manuscript by assigning *Cutibacterium acnes* to *Propionibacterium* genus during the evaluation for the HMP dataset. The new result increased the genus level precision from 0.50 to 0.64, with the weighted recall only dropping 0.005 (from 0.691 to 0.686). We again note that genera incorporates considerable spp delineations, and do not suggest that all spp from this genera ought to be considered contaminants in human studies.

The following text has been updated.

> (Results, Genus level accuracy of Squeegie prediction, Page 3)

For the HMP dataset at genus rank, Squeegie achieved a precision of 0.64 (16/25 genera) and a recall of 0.262 (16/61 genera). Although only 16 genera were correctly predicted, those genera accounted for the majority of the contaminated reads in the ground truth with the total relative abundance of 68.6%.

> (Results, Species level accuracy of Squeegie prediction, Page 5)

Squeegie achieved a precision of 0.813 (61/75 species).

> (Result, Alpha diversity analysis before and after contamination removal, Page 7)

Figure 6b shows the same alpha diversity analyses performed on the HMP samples with the maximum removal cutoff set to 1%. We observed significant decreases for Shannon's diversity index values in oral and nasal samples, and a significant decrease in Simpson's diversity index in oral samples. With the max removal cutoff of 0.5%, there are significant decreases of Shannon's diversity index and Simpson's diversity index in the oral samples (See Supplementary Fig.4).

> (Discussion, Page 9)

Over 81% of the contaminant species predicted by Squeegie for the HMP dataset match the bacterial genus described as inherent contaminants of the MoBio DNA extraction kit, which was used for the Human Microbiome Project. The Squeegie prediction has a weighted recall of 0.686, and Squeegie failed to predict most of the genera from phylum.

> (Methods, Evaluation of Squeegie, Page 11-12)

For (3) the HMP dataset, which was extracted using the MoBio DNA extraction kit, we used the 61 bacteria genera (excluding lot dependent organisms) which were identified as inherent contaminants within a latter version of a related MoBio extraction kit, the MoBio PowerMax Soil DNA Isolation Kit 12,988-10 (MoBio Laboratories, USA), in a recent study as the ground truth contaminants.

> (Methods, Evaluation of Squeegie, Page 12)

During the evaluation, *Cutibacterium acnes* (formerly *Propionibacterium acnes*) was assigned to genus *Propionibacterium* to keep the ground truth and Squeegie prediction consistent.

The following display items have been updated.

Figure 2, Please refer to R1Q1

Figure 3. Relative abundance of ground truth contaminants of (a) maternal/infant and (b) HMP dataset. The correct predicted species/genera are marked in blue, and the species/genera that Squeegie failed to predict is marked in red. For HMP dataset, genera with relative abundance below 1% are combined.

Figure 6. Alpha diversity index for (a) maternal/infant dataset and (b) HMP dataset. Both Shannon's and Simpson's diversity index of the communities in each of the samples were evaluated before the contaminant reads were removed (red), after removing species only confirmed by the experimental negative control (blue), and after removing all species predicted by Squeegee (black). The max removal is set to 1%. Numbers inside parentheses are the numbers of samples in each sample type. Significance labeling: n.s.($P > 0.05$), *($P \leq 0.05$), **($P \leq 0.01$), ***($P \leq 0.001$). Each box plot includes median line, and the box bounds the interquartile range (IQR). The Tukey-style whiskers extend from the box by at most $1.5 \times$ IQR. The circle denotes outliers that extend beyond the whiskers.

Supplementary Figure 1. Scoring and filtering of candidate contaminants for the HMP dataset. This plot shows the prevalence, the breadth of genome coverage, and additional score and filtering information of the top 50 contaminant species after filtering. The first 16 rows show the prevalence of each species among each of the sample types, where zero prevalence is marked in blue. The next 16 rows show the breadth of genome coverage of each species in each of the sample types, where zero is marked in blue. The remaining rows show the prevalence score, the alignment score, the Mash score, and the combined score used to make the final prediction, and whether each species passes the filters. The last row of the heat map shows whether the species can be found in the ground truth with true positive show in white and false positive show in black.

Supplementary Figure 4. Alpha diversity index for HMP dataset. Both Shannon's and Simpson's diversity index of the communities in each of the samples were evaluated before the contaminant reads were removed (red), after removing species only confirmed by the experimental negative control (blue), and after removing all species predicted by Squeegee (black). The max removal is set to 0.5%. Numbers inside parentheses are the numbers of samples in each sample type. Significance labeling: n.s.($P > 0.05$), *($P \leq 0.05$), **($P \leq 0.01$), ***($P \leq 0.001$). Each box plot includes median line, and the box bounds the interquartile range (IQR). The Tukey-style whiskers extend from the box by at most $1.5 \times$ IQR. The circle denotes outliers that extend beyond the whiskers.

	Simulated	Maternal/infant	HMP	Human RNA-Seq
Total number of samples	30	344	749	40
Total number of sample types	6	9	16	6
Prevalence min read	30	30	30	30
Prevalence min abundance	0.05%	0.05%	0.05%	0.05%
Min genome coverage	7.5%	2.5%	7.5%	1.0%
Min combined score	0.75	0.75	0.75	0.75
# of ground truth species	12	16	N/A	N/A
# of ground truth genus	9	14	61	N/A
# of predicted species	11	14	75	7
# of predicted genus	7	12	25	6
# of correct predicted species	8	10	61	N/A
# of correct predicted genus	5	10	16	N/A

Supplementary Table 1. Parameters and Dataset Characteristics

R2Q6. Squeegie combines multiple features to a single score to determine if some species is a contaminant or a real signal. How much does each of these terms contribute to the prediction performance? What is the range of the score? How does the score relate to precision or sensitivity? Providing a fitting from the score to sensitivity/precision would be a great help for users to set this complex parameter.

Squeegie uses a dynamic scoring mechanism to favor more distinguishable features while making the final prediction. We have added detailed descriptions for each of the scores, as well as how parameter setting affects the false positive and false negative. Additional discussion on how to choose the suitable parameter based on different sample characteristics has been added to the manuscript.

For additional details, we refer the reviewer to responses to R1Q2 and R1Q6.

R2Q7. How does contamination in the DNA database affect Squeegie? Many pathogen genomes contain human DNA contaminants. If performing metagenomic pathogen detection in humans then many pathogen genomes will occur in samples. However, this might be due to human DNA in their genomes. Would these pathogens be labeled as contaminants?

Not likely. Squeegie takes sequencing reads as input to determine contamination in the samples. The user should perform human read removal to remove human sequences from the input, as described in the methods section. Assuming the human reads are correctly removed from the input, if a pathogen genome is contaminated with the human genome, the breadth of genome coverage for that pathogen would be underestimated since the contaminated region would have no coverage.

As a result, using a database with pathogen genomes containing human DNA contaminants, and depending on what proportion of the pathogen genomes is contaminated, the recall would potentially decrease.

R2Q8. Typo in Figure 1 "Contamiant_score" should be "Contaminant_score"

The typo in Figure 1 has been corrected.

R2Q9. Please mention the conda package in the manuscript and cite bioconda.

Citations for bioconda have been added to multiple places in the manuscript. The following article has been added to the reference.

> (Reference, Page 14)

Grüning, B. et al. Bioconda: sustainable and comprehensive software distribution for the life sciences. Nat. methods 15, 475–476 (2018).

R2Q10. Metagenomic samples can be large and run time can be critical for users. Please discuss the resource and runtime requirements of Squeegiee.

We have added a set of experiments to test the run time and memory usage of Squeegiee.

> (Results, Run time and memory usage, Page 7)

We tested the performance of Squeegiee with datasets of different sizes. Table 1 shows the run time and peak memory usage of Squeegiee for different sizes of input. The run time of Squeegiee (in CPU hours) is largely determined by the size of the input data, and is also affected by the number of potential contaminants identified based on the taxonomic classification results. Since the reads are mapped to reference genomes of each potential contaminant using Bowtie2 (with multi-alignments enabled), more contaminants would increase the alignment CPU time as well as the coverage calculation process. The peak memory usage of Squeegiee is largely driven by the size of the database used by Kraken for taxonomic classification. The following runs are using the same Kraken database (302 GB) built with NCBI RefSeq (Release 202), thus they have similar peak memory usage.

Table 1. Run time and memory usage

	Test set 1	Test set 2	Test set 3
Input Size (GB)	59.2	59.2	118.0
# of Potential Contaminants	15	18	26
CPU hours	55.5	63.3	130.3
Peak Memory Usage including Kraken (GB)	320.4	320.4	322.0

R2Q11. Please open source all scripts and data to make the study reproducible

The data availability statement and code availability statement have been added to the manuscript.

> (Data availability, Page 12)

The source data generated in this study has been deposited in the OSF database with <https://osf.io/ap7cd/>. All sequencing data supporting the findings of this study is publicly available. The simulated dataset is publicly available and can be downloaded at <https://zenodo.org/record/5841968>. The maternal/infant metagenomic datasets are available for download via NCBI BioProject PRJNA725597. The HMP samples are downloaded from <https://www.hmpdacc.org/HMASM/>. The human RNA-Seq datasets are available for download via NCBI BioProject PRJEB2123. The datasets were simulated/sequenced from distinct samples, and no sample was simulated/sequenced repeatedly.

> (Code availability, Page 13)

The source code for Squeegie is publicly available at <https://gitlab.com/treangenlab/squeegie>, and we used version 0.2.0 of Squeegie for the result and analysis presented in this manuscript. The code used for analysis and figure generation used in this study can be found in: <https://osf.io/ap7cd/>.

Reviewer #3 (Remarks to the Author)

The manuscript entitled “Squeegie: de-novo identification of reagent and laboratory induced microbial contaminants in low biomass microbiomes” gives a free computational method “Squeegie”, which can identify potential contaminants, and the manuscript focuses on testing it. But in this article, some problems remain.

R3Q1. This study mainly detects microbial contaminants through “Squeegie”, but the data is from Human Microbiome Project, we expect authors to provide their original metagenomic data to test “Squeegie”.

We thank the reviewer for the feedback. The benchmark of Squeegie contains 3 datasets including a simulated dataset, a real dataset containing low biomass maternal/infant samples, and HMP samples. The maternal/infant dataset is considered as our original metagenomic dataset. It contains 344 samples in total across 9 different sample types. The dataset also has multiple negative control sequenced samples and those negative control samples were used as a ground truth to evaluate Squeegie.

We also added a new experiment with a publicly available human RNA-Seq dataset. For the details of the datasets used in the manuscript, please refer to R2Q5 (Supplementary Table 1. Parameters and Dataset Characteristics)

R3Q2. For the HMP dataset at genus rank, “Squeegie” achieved a precision of 0.5 and a recall of 0.328. We hope that authors can improve “Squeegie” to achieve higher precision and recall.

The original low precision is partially related to an error caused by taxonomic re-assignment of *Cutibacterium acnes* (formerly *Propionibacterium acnes*). We have fixed the issue and redo the analysis on the HMP dataset. The new result indicates Squeegie has a genus level precision of 0.64 (up from 0.5), and a weighted recall of 0.686 (down from 0.691). The recall is 0.262.

The reason for the low recall is that the ground truth contaminants we evaluate Squeegie on contain a large number of low abundant genus. Those low abundant genus have ultra low signals as the true biological samples are being added to the DNA extraction kit, and they are untraceable

with limited sequencing depth. Although only 16 out of 61 genera were correctly predicted, these genera accounted for the majority of the contaminated reads in the ground truth with the total relative abundance of 68.5%. (R2Q5, updated display item, Figure 3)

For additional details about the updated analysis result for the HMP dataset, we refer the reviewer to R2Q5.

R3Q3. Figure 6, Supplementary Fig. 2, and Supplementary Fig. 3 show different results when the max removal cutoff is set to 1% and 0.5%, please explain it. And why is the max removal cutoff set to 0.5% ?

The reason for using a maximum removal cutoff while performing alpha diversity analysis is because of ubiquitous species, which are species that can be found in a wide range of environments. Squeegee predicts contaminants without prior knowledge of the samples, and its calculation is based on the assumption that the contaminant species originated from the same source, such as lab environment and lab preparation kits. The ubiquitous species can be at the same time contaminants from the external source, and true members of the microbiome community that we sampled from. For example, *Staphylococcus sp.* is considered as a ubiquitous species. It has been reported as kit contaminants in multiple previous studies, but indeed it's also a part of the human skin microbiota.

Removing ubiquitous species directly from the sampled community regardless of relative abundance can bias the downstream analysis, such as alpha diversity calculation. So we used a maximum removal cutoff to show that removing highly confident contaminants even at low abundance could still impact alpha diversity index.

Additional Changes

> (Title, Page 1)

Identifying Microbial Contaminants in Low Biomass Microbiomes when Negative Controls are Unavailable

> (Abstract, Page 1)

We also analyzed 749 metagenomic samples of varying biomass with the Human Microbiome Project and identified likely but previously unreported kit contamination. Collectively, our results highlight that Squeegee can identify microbial contaminants with high precision, and represents a valuable computational tool for contaminant detection when negative controls are unavailable.

> (Introduction, Page 1)

Experimental negative and/or environmental contaminant controls combined with computational contamination identification and removal is effective

> (Result, Page 3)

We evaluated Squeegee on 3 datasets, including: (i) a simulated dataset with ground truth contaminants, (ii) a real dataset with negative controls, and (iii) HMP samples without negative controls but with associated DNA extraction kit contaminants. Details on the implementation and evaluation of Squeegee can be found in the methods section.

> (Result, Species level accuracy of Squeegie prediction)

Figure 5 has been moved to Supplementary Figure 2.

> (Result, Species level accuracy of Squeegie prediction, Page 4)

The false positive calls include ~~Cutibacterium acnes~~, *Rothia mucilaginosa*, *Staphylococcus cohnii*, *Staphylococcus mitis*, *Staphylococcus haemolyticus*, which led to a precision of 0.714.

> (Discussion, Page 7)

Squeegie is able to mark these taxa contained within metagenomic samples without requiring negative experimental controls, and can identify potentially widespread contaminants in public data.

> (Discussion, Page 7)

With breadth of genome coverage for each contaminant species being calculated, Squeegie also resolves the the taxonomic classification error that might occur during taxonomic binning process, which is an common issue for k-mer based methods

> (Discussion, Page 7)

In the maternal/infant dataset, Squeegie predicted most of the contaminants found in the negative control samples, including species from contaminating genera (e.g., *Methylobacterium*, *Pseudomonas*, and *Xanthomonas*) that have been previously reported. For the false negative contaminant species the Squeegie failed to predict, they all have relative abundances below 5% except for *Staphylococcus capitis*. In addition to other genera and species unique to the low biomass maternal/infant samples, we also found that Squeegie predicted a number of contaminant species from the genera *Staphylococcus* including *Staphylococcus haemolyticus*, *Staphylococcus mitis*, and *Staphylococcus cohnii*, that are not found in the experimental control samples.

> (Discussion, Page 8)

Squeegie failed to predict some of the low abundance genera/species in the simulation dataset due to similar reasons (see Supplementary Fig. 1). In order to challenge Squeegie, the simulation dataset we designed contains extremely low proportions (0.05%-1%) of spike-in contaminant sequences, and among the 12 spike-in contaminant species, all of them except *Ralstonia pickettii* have relative abundance below 0.1. In order to detect low abundance contaminants, using Decontam with negative control samples is recommended.

> (Discussion, Page 10)

Squeegie predictions on multiple datasets have shown that contaminant sequences from the same source, such as DNA extraction kits and other reagents used during the sample processing and sequencing, can be accurately identified across multiple samples using this computational method without experimental negative controls or DNA quantitation data.

>(Acknowledgements, Page 15)

Special thanks to the depositors of all publicly available metagenomic datasets used in this study.

>(Funding, Page 15)

R.A.L.E. was supported in part by a training fellowship from the Gulf Coast Consortia, on the NLM Training Program in Biomedical Informatics & Data Science (T15LM007093).

>(Funding, Page 15)

This work was supported in part by the Big-Data Private-Cloud Research Cyberinfrastructure MRI-award funded by NSF under grant CNS-1338099 and by Rice University's Center for Research Computing (CRC). Additional support for this work arose from the NIH (NIH-NINR R01NR014792, NIH-NICHHD R01HD091731, both to K.A.).

The following references have been added.

O'Callaghan, J. L. et al. Re-assessing microbiomes in the low-biomass reproductive niche. *BJOG: An Int. J. Obstet. & Gynaecol.* **127**, 147–158 (2020).

Gastauer, M. et al. A metagenomic survey of soil microbial communities along a rehabilitation chronosequence after iron ore mining. *Sci. data* **6**, 1–10 (2019).

Abraham, B. S. et al. Shotgun metagenomic analysis of microbial communities from the loxahatchee nature preserve in the florida everglades. *Environ. microbiome* **15**, 1–10 (2020).

Buck, M. et al. Comprehensive dataset of shotgun metagenomes from oxygen stratified freshwater lakes and ponds. *Sci. Data* **8**, 1–10 (2021).

Mangrola, A. et al. Deciphering the microbiota of tuwa hot spring, india using shotgun metagenomic sequencing approach. *Genomics data* **4**, 153–155 (2015).

Oh, J. et al. Biogeography and individuality shape function in the human skin metagenome. *Nature* **514**, 59–64 (2014).

Mas-Lloret, J. et al. Gut microbiome diversity detected by high-coverage 16s and shotgun sequencing of paired stool and colon sample. *Sci. data* **7**, 1–13 (2020).

Strong, M. J. et al. Microbial contamination in next generation sequencing: implications for sequence-based analysis of clinical samples. *PLoS pathogens* **10**, e1004437 (2014).

t Hoen, P. A. et al. Reproducibility of high-throughput mrna and small rna sequencing across laboratories. *Nat. biotechnology* **31**, 1015–1022 (2013).

Nasko, D. J., Koren, S., Phillippy, A. M. & Treangen, T. J. Refseq database growth influences the accuracy of k-mer-based lowest common ancestor species identification. *Genome biology* **19**, 1–10 (2018).

Grüning, B. et al. Bioconda: sustainable and comprehensive software distribution for the life sciences. *Nat. methods* **15**, 475–476 (2018).

Meyer, F. et al. Critical assessment of metagenome interpretation: the second round of challenges. *Nat. Methods* 1–12 (2022).

Breitwieser, F. P., Baker, D. & Salzberg, S. L. Krakenuniq: confident and fast metagenomics classification using unique k-mer counts. *Genome biology* **19**, 1–10 (2018).

Aagaard, K. et al. The human microbiome project strategy for comprehensive sampling of the human microbiome and why it matters. *The FASEB J.* **27**, 1012–1022 (2013). 14/15

Reviewers' comments:

Reviewer #1 (Remarks to the Author):

In this revision, Liu and colleagues addressed many concerns raised with the previous version of the manuscript. In particular, the fuller description of the method parameters is appreciated. The approach of using a metric of sample dissimilarity (like MASH scores) to weight the evidence for cross-sample contamination remains novel and interesting. However, some important concerns remain.

—

Weighted precision needs to be added as an evaluation metric. In their rebuttal the authors stated that this can't be done because "there is no truth relative abundance associated with the FP predictions, so we are unable to calculate weighted precision". This is not the case. The FP predictions have a true relative abundance within the samples of the study. So, the weighted precision is simply the average fraction of reads in the TP taxa over those in the TP and FP together taxa. A potential source of confusion may be that the authors have been calculating contaminant relative abundances only in negative controls? This wasn't clear to me previously, but seems to be the case on re-reading. If so, that should be changed throughout. The relevant relative abundances are those obtained from the real samples, not the negative controls.

The test datasets require more complete reporting. Supplementary Table 1 is a start, but much of that information, particularly on sample Ns (preferably within sample types as well) should be in the main text. The ST1 info also needs to be augmented with the number of negative controls in each study.

The way in which precision is evaluated in the infant dataset needs to be reconsidered. The authors have chosen a conservative method to establish definite contaminants – "To minimize classification errors, we applied a set of criterion to include a species in the contamination ground truth. Species with relative abundance above 0.5% or more than 3000 reads assigned in more than half of the negative control samples, and species with relative abundance above 10% in a single sample were chosen for inclusion in the ground truth contaminant set." However, that doesn't work when evaluating precision. Many TP contaminant IDs may be misreported as FPs because the "ground truth" set of contaminants is much smaller than it should be due to this conservative approach. There isn't a ground truth in this dataset, and thus the way in which precision in particular is evaluated needs to be revisited. This is hinted at in the Discussion: "Squeegee failed to predict some of the low abundance genera/species in the simulation dataset ... In order to detect low abundance contaminants, using Decontam with negative control samples is recommended.". Perhaps explore the intersecting and non-intersecting contaminant calls by Squeegee/Decontam in more detail? Because the differing precision scores being reported could be entirely explained by errors in defining the ground truth.

In summary, although the revised manuscript has shown improvement over the original, material issues still remain in the reporting and evaluation of this novel method.

Reviewer #2 (Remarks to the Author):

Thank you for addressing all my points. Congratulations on the manuscript.

LEGEND

Author responses to editor/reviewers in **bold black**

Changes to the manuscript in **blue**

Location of inserted text in (parenthesis)

Added/updated display items indicated in **red**

Reviewers' comments in black

Reviewer #1 (Remarks to the Author):

In this revision, Liu and colleagues addressed many concerns raised with the previous version of the manuscript. In particular, the fuller description of the method parameters is appreciated. The approach of using a metric of sample dissimilarity (like MASH scores) to weight the evidence for cross-sample contamination remains novel and interesting. However, some important concerns remain.

R1Q1. Weighted precision needs to be added as an evaluation metric. In their rebuttal the authors stated that this can't be done because "there is no truth relative abundance associated with the FP predictions, so we are unable to calculate weighted precision". This is not the case. The FP predictions have a true relative abundance within the samples of the study. So, the weighted precision is simply the average fraction of reads in the TP taxa over those in the TP and FP together taxa. A potential source of confusion may be that the authors have been calculating contaminant relative abundances only in negative controls? This wasn't clear to me previously, but seems to be the case on re-reading. If so, that should be changed throughout. The relevant relative abundances are those obtained from the real samples, not the negative controls.

We thank the reviewer for the feedback and raising this important point. To address the reviewer's comment, we have now calculated weighted precision based on the true relative abundance of the taxa within the non-control samples of the studies. In addition to the updated weighted precision, we added the weighted recall and weighted F-score both calculated based on the true relative abundance of the taxa within the non-control sample as suggested by the reviewer. The details of these calculations are as follows:

- **The weighted precision is calculated as the average fraction of reads from the non-control samples in the true positive (TP) taxa over those in the true positive and false positive (TP+FP) taxa, at both genus and species rank.**
- **The weighted recall is calculated as the average fraction of reads from the non-control samples in the true positive (TP) taxa over those in the true positive and false negative (TP+FN) taxa, at both genus and species rank.**
- **The weighted F-score is calculated as $2 * (\text{weighted precision} * \text{weighted recall}) / (\text{weighted precision} + \text{weighted recall})$**

We feel that the relative abundance of the contaminants in the negative control samples still serves as an important measurement of how well the methods perform, since the negative experiments reflect the contaminants and potential bias contribution of different contaminants to the downstream analysis. We kept the weighted recall measurement we used in the previous version of the manuscript as ‘weighted by relative abundance in negative controls’ and showed a detailed composition of the contaminant taxa in the following updated figures.

Please note that the ground truth contaminant composition has also been updated based on the suggestion of the reviewer in R1Q3, and per the reviewers feedback, we have added more permissive criteria to establish the ground truth contaminants. In combination with the new evaluation measurements, we updated our results as following:

The following display items have been updated in accordance with the reviewers feedback:

a) Evaluated with the permissive ground truth contaminant set

b) Evaluated with the strict ground truth contaminant set

Figure 2. Squeezegee (de novo) and Decontam (with negative control) accuracy at species and genus ranks evaluated with (a) the permissive ground truth and (b) the more strict ground truth. The figures show the precision, recall, and F-score calculated at species and genus rank for both methods. The unweighted precision is calculated as the ratio between the number of predicted contaminant taxa found in the ground truth and the total number of predicted contaminant taxa. The unweighted recall is calculated as the ratio between the number of predicted contaminant taxa found in the ground truth and the total number of taxa in the ground truth. While weighted by samples, the measurements are weighted by the mean proportion of the reads assigned to each taxa in the non-control experiment samples. The weighted by negative controls figures show the detailed composition of the taxa and their mean relative abundance in the negative control samples, and the cumulative relative abundance of the correctly predicted putative contaminants (weighted recall) by different methods. The correctly predicted species/genera are marked with stripes, and the species/genera that the methods failed to predict are without stripes. Multiple low relative abundance taxa have been combined in Figure (a).

Figure 3. Squeegee performance on HMP metagenomic datasets. (a) Left panel depicts the Genus level precision, recall, and F-score using previously reported kit contaminants as the ground truth. Unweighted precision is calculated as the ratio between the number of predicted contaminant taxa found in the ground truth and the total number of predicted contaminant taxa. Unweighted recall is calculated as the ratio between the number of predicted contaminant taxa found in the ground truth and the total number of taxa in the ground truth. While weighted by samples, the measurements are weighted by the mean proportion of the reads assigned to each taxa in the non-control experiment samples. (b) The right panel highlights the correctly predicted genera are marked in orange with stripes, and the genera that Squeegee failed to predict are marked in gray. Genera with relative abundance below 1% are combined.

Furthermore, the following text has been updated:

> (Results, Benchmark with Decontam, Page 3-5)

We evaluated Squeegee prediction accuracy at both genus and species rank on the maternal/infant datasets. During this benchmark, Squeegee performed contamination prediction without the use of the negative control samples, while Decontam took the classification results of the 10 negative control samples as input for contamination identification. A permissive ground truth contaminant set, and a strict version of the ground truth contaminant set, are generated with data from the negative control samples as well to use as reference for the evaluation, where the strict set is generated with more stringent filtering to ensure high confidence. The details of the contaminant ground truth sets can be found in the methods section.

Figure 2a shows the precision, recall, and F-score of Squeegee and Decontam at both species and genus rank using the permissive ground truth set. The unweighted precision, unweighted recall, and unweighted F-score for Squeegee are 0.714 (10/14 species), 0.323 (10/31 species) and 0.444 at species rank, and 0.833 (10/12 genera), 0.625 (10/16 genera), and 0.714 at genus rank, respectively. The false positive calls for Squeegee are: *Rothia mucilaginosa*, *Staphylococcus cohnii*, *Staphylococcus haemolyticus* and *Streptococcus mitis*. The unweighted precision, unweighted recall, and unweighted F-score for Decontam are 0.140, 0.774, and 0.238 at species rank, along with 0.174, 0.750, and 0.282 at genus rank, respectively.

We also evaluated both methods with weighted scores, taking into account abundance information. Each of the species are first weighted by the mean fraction of reads assigned to those species in the non-negative samples. The weighted precision, weighted recall, and weighted F-score for Squeegee is 0.580, 0.728, and 0.645, respectively, and for Decontam is 0.928, 0.494, and 0.645, respectively, at species rank. The same measurements at genus rank were 0.438, 0.804, and 0.567 for Squeegee, respectively, and 0.947, 0.732, and 0.826 for Decontam, respectively.

More importantly, we took a closer look at the predicted contaminants output by each of the methods, and evaluated the recall weighted by the relative abundance of the taxa in the negative control samples. Although Squeegee failed to identify several of the putative low abundance contaminant species, the 10 correctly predicted species by Squeegee occupy over 0.763 of the cumulative relative abundance from the composition of the putative ground truth contaminants. Using the same measurement, the species rank weighted recall under the same criteria for Decontam is 0.645. At genus rank, both methods performed well, with weighted recall for Squeegee scored at 0.892 and Decontam scored at 0.921. Although Decontam mislabeled some of the high abundance contaminants at species rank, it did label some of the closely related species under the same genera as contaminants, resulting in a significant

increase of the score at genus rank. Figure 2b shows the accuracy of Squeegie and Decontam using the strict ground truth set. The detailed results can be found in the Supplementary Section 1.

> (Results, Accuracy of Squeegie on the HMP datasets, Page 5)

We evaluated Squeegie prediction accuracy on the HMP datasets as well. Figure 4 shows the precision, recall, and F-score of Squeegie predictions at genus rank. Squeegie has an unweighted precision of 0.667 (16/24 genera), an unweighted recall of 0.262 (16/61 genera), and an unweighted F-score of 0.376. While each of the taxa is weighted by their relative abundance from the non-control samples, Squeegie achieved a weighted precision of 0.856, a weighted recall of 0.958, and resulted in a weighted F-score of 0.904. Figure 4 also shows the relative abundance of true contaminant genera identified in the MoBio DNA extraction kit. The contaminants successfully predicted by Squeegie are colored in orange with stripes and the contaminants Squeegie failed to predict are colored in gray. Low abundance genera with relative abundance below 1% are combined in the figure. Although only 16 genera were correctly predicted, those genera accounted for the majority of the contaminated reads in the ground truth with the total relative abundance of 0.686.

> (Results, Accuracy of Squeegie on the HMP datasets, Page 5)

Since we are using bacteria identified at the genus level as inherent putative contaminants in the MoBio DNA extraction kit level for our negative control reference, accuracy measurements at the species level do not apply. It is worth noting that more than 81.3% (61 out of 75) species that Squeegie predicted as contaminant species in the HMP datasets fell under the ground truth contaminant genera.

> (Discussion, Page 8-9)

We benchmarked Squeegie against a “gold-standard” contamination detection approach in Decontam, with its prevalence-based method requiring negative control samples as input. We note that we view both tools as complementary, especially since the use of negative controls are recommended best practice for contamination removal. From the results (Figure 2b) we see that Squeegie is able to achieve performance that meets or exceeds Decontam predictions at species rank using the strict ground truth, with respect to unweighted F-score, weighted F-score by the relative abundance in the non-control samples, and cumulative relative abundance of the putative correctly identified contaminants from the negative control experiment samples. On the other hand, at the genus rank, Squeegie is unable to match Decontam performance when F-score is weighted by the relative abundance in the non-control samples, while performing on par with Decontam with respect to cumulative relative abundance of the correctly identified contaminant genera in negative controls. Although Decontam failed to recognize *Pseudomonas tolaasii* and *Xanthomonas euvesicatoria* as putative contaminant species, multiple species under the same genera were successfully identified, increasing its genus rank score. While evaluating using the permissive ground truth contaminant set, Squeegie performed equally well at species rank with respect to weighted F-score (Figure 2a), with a drop in unweighted recall given Squeegie failed to recognize most contaminant species with mean relative abundance less than 1% in the negative controls.

> (Discussion, Page 10)

We looked closer into the low weighted precision of non-control sample abundance at species and genus rank for Squeegie in the maternal/infant dataset. *Streptococcus mitis*, a common member of the microbial communities from oral, skin, the female genital tract, and gastrointestinal tract, was incorrectly identified as a contaminant. Given the high relative abundance of *Streptococcus mitis* (Figure 4), this false positive contaminant species received abnormally high weight compared to the other true positive contaminant species, lowering Squeegie's weighted precision. At genus rank, the relative abundance difference between *Streptococcus* and the true positive putative contaminant genera becomes even greater, which explains why Squeegie genus rank performance is lowered compared to species rank in

this dataset. These results highlight areas for future improvement to Squeegie that would allow it to take into account microbial species ubiquitous to many different environments.

> (Discussion, Page 11)

Squeegie achieves high weighted recall (weighted by both relative abundance of taxa in negative control and non-control samples) and low false positive rates on real metagenomic datasets,...

> (Methods, Evaluation of Squeegie, Page 14)

Performance is evaluated via precision, recall, and F-score. The unweighted precision is calculated as the ratio between the number of predicted contaminants found in the ground truth and the total number of predicted contaminants. The unweighted recall is calculated as the ratio between the number of correctly predicted contaminants and the total number of contaminants in the ground truth. The unweighted F-score is calculated as $2 * (\text{unweighted precision} * \text{unweighted recall}) / (\text{unweighted precision} + \text{unweighted recall})$. Those three measurements are also calculated using the mean fraction of the reads of each taxon in the non-control samples as weight. The weighted precision is calculated as the average fraction of reads from the non-control samples in the true positive (TP) taxa over those in the true positive and false positive (TP+FP) taxa. The weighted recall is calculated as the average fraction of reads from the non-control samples in the true positive (TP) taxa over those in the true positive and false negative (TP+FN) taxa. The weighted F-score is calculated as $2 * (\text{weighted precision} * \text{weighted recall}) / (\text{weighted precision} + \text{weighted recall})$. We also evaluated the methods using cumulative relative abundance of true positive taxa, which is a weighted recall score that is weighted by the mean relative abundance of the taxa in the negative control samples.

> (Supplementary Section 1, Benchmark with Decontam using the strict ground truth, Page 1)

Figure 2b shows the precision, recall, and F-score of Squeegie and Decontam at both species and genus rank using the strict ground truth set. At species rank, Squeegie achieved an unweighted precision of 0.643 (9/14 species) and an unweighted recall of 0.600 (9/15 species). The false positive calls for Squeegie are: *Rothia mucilaginos*a, *Staphylococcus cohnii*, *Staphylococcus mitis*, *Staphylococcus haemolyticus*, and *Escherichia coli*. The unweighted F-score for Squeegie is 0.621. On this dataset, Decontam achieved an unweighted precision of 0.053, an unweighted recall of 0.600, and an unweighted F-score of 0.097.

We also evaluated both methods with weighted scores, taking into account abundance information. Each of the species are weighted by the mean fraction of reads assigned to those species in the non-negative samples. The weighted precision, weighted recall, and weighted F-score for Squeegie is 0.502, 0.835, and 0.627, and for Decontam is 0.651, 0.459, and 0.538. We also took a closer look at the predicted contaminants output by each of the methods, and their relative abundance in the negative control samples. The 9 correctly predicted species by Squeegie occupy over 0.835 of the cumulative relative abundance from the composition of the ground truth contaminants. Using the same measurement, Decontam identifies 0.616 of the cumulative relative abundance from the ground truth species.

At genus rank, while evaluating using the strict ground truth set, Squeegie achieved an unweighted precision of 0.750 (9/12 genera) and an unweighted recall of 0.692 (9/13 genera), and resulted in an unweighted F-score of 0.720. Decontam achieved an unweighted precision of 0.145, an unweighted recall of 0.769, and an unweighted F-score of 0.244. When evaluating both methods by weighting each genus by the mean fraction of reads of all genera found in the non-negative samples. The weighted precision, weighted recall, and weighted F-score for Squeegie was 0.391, 0.897, and 0.545, respectively, and the weighted precision, recall, and F-score of Decontam was 0.800, 0.773, and 0.787, respectively. When

evaluated with recall weighted by the relative abundance of the predicted genera in the negative control samples, both methods performed well, with Squeegee scored at 0.940 and Decontam scored at 0.922. Although Decontam mislabeled some of the high abundance contaminants at species rank, it did label some of the closely related species under the same genera as contaminants, resulting in a significant increase of the score at genus rank.

R1Q2. The test datasets require more complete reporting. Supplementary Table 1 is a start, but much of that information, particularly on sample Ns (preferably within sample types as well) should be in the main text. The ST1 info also needs to be augmented with the number of negative controls in each study.

We thank the reviewer for the comment. We have added the following text in the main text describing the sample Ns and number of sample types.

> (Methods, Evaluation of Squeegee, Page 13)

The simulated dataset contains a total of 126 simulated samples representing 7 distinct microbial communities. The real dataset contains 344 samples over 9 distinct sample types collected of adult females and infants, as well as sequencing data from 10 negative control experiment samples. The HMP dataset includes 749 samples collected and sequenced from health individuals across 16 different body sites.

Supplementary Table 1 is now updated with the number of negative controls in each study. Please refer to R1Q3. We also have prepared a manifest file that lists each of the samples used in this manuscript.

R1Q3. The way in which precision is evaluated in the infant dataset needs to be reconsidered. The authors have chosen a conservative method to establish definite contaminants – “To minimize classification errors, we applied a set of criterion to include a species in the contamination ground truth. Species with relative abundance above 0.5% or more than 3000 reads assigned in more than half of the negative control samples, and species with relative abundance above 10% in a single sample were chosen for inclusion in the ground truth contaminant set.” However, that doesn’t work when evaluating precision. Many TP contaminant IDs may be misreported as FPs because the “ground truth” set of contaminants is much smaller than it should be due to this conservative approach. There isn’t a ground truth in this dataset, and thus the way in which precision in particular is evaluated needs to be revisited. This is hinted at in the Discussion: “Squeegee failed to predict some of the low abundance genera/species in the simulation dataset ... In order to detect low abundance contaminants, using Decontam with negative control samples is recommended.”. Perhaps explore the intersecting and non-intersecting contaminant calls by Squeegee/Decontam in more detail? Because the differing precision scores being reported could be entirely explained by errors in defining the ground truth.

We thank the reviewer for raising these points. We will now address each of the points, and also explore in detail the intersecting and non-intersecting Squeegee/Decontam calls. Since k-mer based classification methods are prone to noisy/erroneous assignments for the low abundance species, our goal was to create a conservativate method for establishing definite contaminants to minimize the impact of taxonomic classification errors (which we

consider to be classifier "contamination") while maintaining what we felt was a reasonable ground truth set. However, we understand the reviewers' concern with respect to misreporting TP contaminant IDs as FPs.

Thus, in response to the reviewer's concern, we have completely revamped the evaluation in the current version of the manuscript, we changed the criteria to identify a species as a true member of the contaminants, and also added a permissive ground truth set.

- The strict criteria for a species to be identified as a true member of the contaminant set is as follows:
 - Species with relative abundance equal or above 0.5%, and having equal or more than 20 reads assigned in at least half of the negative control samples, plus species with relative abundance above 10% in a single sample were chosen for inclusion in the permissive ground truth contaminant set.
- The permissive criteria for a species to be identified as a true member of the contaminant set is as follows:
 - Species with relative abundance equal or above 0.2%, and having equal or more than 20 reads assigned in at least half of the negative control samples, plus species with relative abundance above 10% in a single sample were chosen for inclusion in the permissive ground truth contaminant set.
- The only difference between the strict ground truth set and the permissive ground truth set is the relative abundance threshold used, per the reviewer's feedback.

Under these new criteria, the permissive ground truth contaminants becomes a set of 31 species (strict set: 15) across 16 genera (strict set: 13), and almost doubles the number of contaminant species compared to the previous evaluation. The new ground truth contains 17 low relative abundance species with relative abundance lower than 1%.

For the new evaluation results with the updated ground truth contaminant composition on the maternal/infant datasets, please refer to R1Q1.

> (Methods, Evaluation of Squeegiee, Page 13)

Species with relative abundance above 0.5% and more than 20 reads assigned in at least half of the negative control samples, and species with relative abundance above 10% in a single sample were chosen for inclusion in the ground truth contaminant set, resulting in a strict ground truth set with 15 species from 13 genera. We also applied more permissive filtering and generated a permissive ground truth contaminant set that contains 31 species that belongs to 16 genera, by lowering the minimum relative abundance threshold to 0.2%.

> (Discussion, Page 8)

For the maternal/infant dataset, a strict contaminant ground truth and a permissive contaminant ground truth were constructed with the taxonomic assignment of the sequencing data from the negative control experiments with the use of prevalence, relative abundance, and absolute read count filtering, a common practice to minimize the taxonomic assignment error and determine the presence or absent of species, with different filtering parameters. Squeegiee predicted most of the putative contaminants found in the strict ground truth contaminant set (see Figure 2b).

> (Discussion, Page 9)

As expected, Decontam with the experimental negative control data performs best in terms of unweighted recall (see Figure 2b). Decontam identified 24 out of 31 species within the permissive contaminant ground truth set, only missing *Pseudomonas tolaasii*, *Xanthomonas euvesicatoria*, *Cupriavidus oxalaticus*, *Staphylococcus aureus*, *Pasteurella multocida*, *Klebsiella pneumoniae*, and *Escherichia coli*. It is possible that Decontam did not flag some species as contaminants that are shared between source of contamination and the sampling environment, such as *Staphylococcus aureus* and *Escherichia coli*. At the same time, Squeegee identified all those 7 species, which the relative abundance in the permissive ground truth adds up to 35.5%, as contaminants, which completes the entire contaminant ground truth set if we take the union of the predictions made by the two methods. Alternatively, we acknowledge that this may over-call "contamination" by virtue of shared species among body niches. This once again highlights the complementarity of Decontam with negative controls and Squeegee, and also the value of Squeegee either when negative controls are unavailable (existing metagenomic sequence datasets) and for lab contamination that affects both the negative control and samples.

The following display items have been updated.

> (Supplementary Table 1. Parameters and Dataset Characteristics)

Supplementary Table 1. Parameters and Dataset Characteristics

	Simulated	Maternal/infant	HMP	Human RNA-Seq
Total number of samples	126	344	749	40
Total number of sample types	7	9	16	6
Prevalence min read	30	30	30	30
Prevalence min abundance	0.05%	0.05%	0.05%	0.05%
Min genome coverage	7.5%	2.5%	20%	1.0%
Min combined score	0.75	0.75	0.75	0.75
# of negative control experiment samples	N/A	10	N/A	N/A
# of ground truth species	12	strict/permissive: 15/31	N/A	N/A
# of ground truth genus	9	strict/permissive: 13/16	61	N/A
# of predicted species	0.25%/0.5%/1%: 5/6/8	14	75	7
# of predicted genus	0.25%/0.5%/1%: 3/4/6	12	24	6
# of correct predicted species	0.25%/0.5%/1%: 5/6/8	strict/permissive: 9/10	61	N/A
# of correct predicted genus	0.25%/0.5%/1%: 3/4/6	strict/permissive: 9/10	16	N/A

Additionally, the reviewer raised the concern that Squeegee does not handle low abundance contaminants very well, since low abundance contaminants have extremely low presence in the actual non-control samples. In order to rigorously evaluate the detection limits of Squeegee, we re-designed the simulation experiments with different levels of spike-in contaminants (0.25%, 0.5%, and 1%), and more realistic non-control sample composition for multiple different environments.

42 real world metagenomic samples from 7 distinct environments are selected, classified, filtered, and used as a reference for the simulation. In comparison with our previous simulation experiments, multiple changes have been made to challenge Squeegee even more:

- For each distinct environment, multiple real world metagenomic samples are used as reference to simulate multiple samples under the same sample type category, creating diversity between simulated samples within the same sample type.

- The number of samples in each sample type are different from each other, demonstrating Squeegee has the capability to recognize the imbalance data, and weight each of the samples accordingly.
- There are three independent runs with different relative abundance of spike-in contaminants, showing that as the spike-in abundance increases, the accuracy of Squeegee increases since there are more signals to identify potential contaminants.
- The total number of simulated samples went from 30 to 126.

The following text has been changed from the manuscript:

> (Methods, Evaluation of Squeegee, Page 13)

In order to simulate a realistic dataset and test the detection limit of Squeegee, 42 real world metagenomic samples were chosen from 7 distinct environments, including 6 soil samples of mining sites, 1 soil sample collected from wetland, 6 freshwater samples, 7 hot spring samples, 6 skin samples of cows, 10 healthy human skin samples, and 6 healthy human gut samples. We filtered out the species with relative abundance lower than 0.0005 or with read support less than 300 in those samples, and used the remaining species and their relative abundance as a reference to simulate the dataset. We then used the species with relative abundance greater than 0.01 found in the FastDNA SPIN Kit for Soil (MP Biomedicals) from the previous study to simulate contaminant sequences. Each distinct sample was simulated 3 times with spiked-in contaminant sequences that occupy 0.25%, 0.5%, 1% of the total sequences in the sample. A total of 126 simulated samples were generated using CAMISIM and ART simulating Illumina paired-end reads with average read length of 150bp and average read pair count of 6664348. The simulated samples are grouped into 3 groups by the spike-in contaminant level, and we evaluate Squeegee on each of the groups individually.

> (Results, Accuracy of Squeegee on the simulated datasets, Page 5-6)

To test the contamination limit of detection of Squeegee, we designed a set of simulated datasets that are based on the taxonomy profile of the real world metagenomic samples. 126 samples are simulated and divided into 3 groups based on different relative abundance of the spike-in contaminant sequences (0.25%, 0.50%, 1.00%). There are 42 simulated datasets in each of the groups, representing microbial communities from 7 distinct environments. The details of how those simulated datasets are generated can be found in the methods section.

> (Results, Accuracy of Squeegee on the simulated datasets, Page 6-7)

Figure 5 shows the unweighted precision, recall, and F-score of different simulated sample groups at species rank, and the same measurement weighted by relative abundance of the taxa in the non-control samples. The figure also shows the detailed composition of the taxa and their relative abundance in the spike-in contaminant community, and the cumulative relative abundance of the correctly predicted contaminants at different relative abundance of spike-in. At all three different spike-in levels, where contaminant sequences occupied 0.25%, 0.50%, and 1.00% of the total reads, Squeegee had the perfect precision of 1.0. For unweighted recall, 0.25% spike-in group scored 0.500, 0.5% spike-in group scored 0.583, and 1.0% spike-in group scored 0.750. As a result, the unweighted F-score for the 0.25% group, 0.50% group, and 1.00% group are 0.667, 0.737, and 0.857. When each species is weighted by their relative abundance in non-control samples, 0.25% spike-in group scored 0.993, 0.5% spike-in group scored 0.990, and 1.0% spike-in group scored 0.989 for the weighted recall, and 0.25% spike-in group scored 0.997, 0.5% spike-in group scored 0.995, and 1.0% spike-in group scored 0.994 for the weighted F-score.

> (Results, Accuracy of Squeegie on the simulated datasets, Page 7)

When each species is weighted by their relative abundance in the negative control, the cumulative relative abundance of true positive prediction for the 0.25% spike-in group is 0.634. As the spike-in level increases, at 0.5% spike-in abundance, Squeegie scored 0.7 with one additional species *Salmonella enterica* identified as contaminant. The cumulative relative abundance of true positive predictions continued to increase at 1.0% spike-in abundance level, and Squeegie scored 0.844 with two more correct contaminant species predicted. In general, the unweighted recall and the cumulative relative abundance of the true positive predictions increases as the number of spike-in contaminant sequences increases, since more contaminant sequences provide stronger signal for Squeegie to pick up on and to make definite calls with respect to contamination.

> (Discussion, Page 9)

In order to challenge Squeegie, the simulation dataset we designed contains extremely low proportions (0.25%-1%) of spike-in contaminant sequences, and among the 12 spike-in contaminant species, all of them except *Ralstonia pickettii* have relative abundance below 0.1. As the relative abundance of the total spike-in sequences increase, we observed that the unweighted recall increases as well, and Squeegie is able to pick up more and more contaminant species. As shown in our experimental results (both simulated and real), Squeegie can exhibit low recall on low abundance contaminant species, which means there will be residual reads not able to be characterized by Squeegie (e.g. they could either represent microbial contaminants or bonafide metagenomic signal).

Figure 5. Squeegie prediction accuracy at species ranks on the simulated datasets. (a) The leftmost panel shows the precision, recall, and F-score calculated at species rank for different relative abundance of spike-in contaminants. The unweighted precision is calculated as the ratio between the number of predicted contaminant taxa found in the ground truth and the total number of predicted contaminant taxa. The unweighted recall is calculated as the ratio between the number of predicted contaminant taxa found in the ground truth and the total number of taxa in the ground truth. **(b)** The center panel shows the same measurements weighted by the mean proportion of the reads assigned to each taxa in the non-control simulated samples. **(c)** The right figure shows the detailed composition of the taxa and their relative abundance in the spike-in contaminant community, and the cumulative relative abundance of the correctly predicted contaminants at different relative abundance of spike-in. The correctly predicted species are marked with striped lines, and the species that Squeegie failed to predict are without striped lines.

Additional Changes

> (Figure 4)

There is no context change in the figure, except the coloring for the TP and FP is now more colorblind friendly. The caption of the figure has been changed to indicate which ground truth was used to generate the figure.

Figure 4. Relative abundance of all predicted species in the maternal/infant dataset. The samples are clustered by their sample type, which is shown with different colors on the color label on the y-axis. The predicted contaminant species that are in the permissive ground truth contaminant set are marked by

the **black** label at the top of the figure (x-axis), whereas the predicted contaminant species that are not found in the strict ground truth contaminant set are indicated in **gray**.

> (Title, Page 1)

Squeegee: Identifying Contaminants in Low Microbial Biomass Microbiomes when Negative Controls are Unavailable

> (Abstract, Page 1)

On the low biomass samples, we compared Squeegee predictions to experimental negative control data and show that Squeegee accurately recovers **putative** contaminants.

> (Introduction, Page 1)

External sources include **personnel**, the laboratory environment, and kits and reagents used for collecting and processing samples. Internal **sources of contamination may include human error, such as sample mislabeling or inadvertent mixing**.

> (Introduction, Page 1)

Studies have shown that contaminants in DNA extraction kits are ubiquitous, and can **bear impact** on metagenomic studies, especially for low-biomass environments, **if** they are not accounted for in the analysis.

> (Introduction, Page 1)

For example, in a recent nasopharyngeal microbiota study on new born babies conducted in Thailand, contaminants found in DNA extraction kits **resulted in contaminant bias**.

> (Introduction, Page 2)

For example, the recently published software Recentrifuge uses a score-oriented comparative approach to identify and remove contaminants from **sequencing reads**.

> (Results, Page 3)

..., **Squeegee identifies** classification errors and make accurate contaminant predictions at the species rank by filtering false calls from the candidates.

> (Supplementary Table 1. Parameters and Dataset Characteristics)

The minimum coverage parameter of the HMP dataset has changed to 20% during previous revision but the number in supplementary table 1 has not been updated. This correction handles the issue. The actual table is shown in R1Q3.

> (Results, **Benchmark with Decontam**, Page 5)

The predicted contaminant species that can be found in the **the permissive ground truth contaminant set** are labeled in **black** at the top of the figure and the predicted contaminant species not found in the **the permissive ground truth contaminant set** are labeled in **light gray**.

> (Results, Alpha diversity analysis before and after contamination removal, Page 7)

Both diversity metrics for the samples were evaluated before the contaminant reads were removed (shown in red), after removing species confirmed by the **permissive ground truth contaminants** (shown in blue).

> (Discussion, Page 8)

A recent study has shown that the accuracy of taxonomic classification algorithms has become a limiting factor of contamination detection, due to high levels of sequence similarity at species rank.

> (Discussion, Page 8)

For the presumptive false negative contaminant species the Squeegie failed to predict, all were of relative abundances below 5% except for *Staphylococcus capitis*.

> (Discussion, Page 8)

..., we also found that Squeegie predicted a number of contaminant species from the genera *Staphylococcus*, including *Staphylococcus haemolyticus*, ~~*Staphylococcus mitis*~~ and *Staphylococcus cohnii*, that are not found in the experimental control samples.

> (Discussion, Page 8)

Additionally, Squeegie called *Rothia mucilaginosa*, which is a part of the normal oropharyngeal flora, and *Escherichia coli*. Both species may represent bonafide species shared across body niches.

> (Discussion, Page 8)

Staphylococcus species are also well-known for their highly similar genomes, which creates a big challenge for the taxonomic assignment task.

> (Discussion, Page 8, the placement of following sentences has been changed)

It is worth pointing out that a stable community member of a certain body site has the potential to also be a contaminant taxon from an external source.

> (Discussion, Page 8)

~~During the benchmark comparing Squeegie and Decontam, we tested Decontam with its prevalence based method, which requires negative control samples as input. Squeegie still outperformed Decontam in both recall and weighted recall at genus and species ranks. It is possible that Decontam did not flag neither those genera nor species as contaminants that are shared between source of contamination and the sampling environment.~~

> (Data availability, Page 15-16)

The simulated datasets are publicly available and can be downloaded at zenodo.org/record/7064705, zenodo.org/record/7062953 and zenodo.org/record/7064599.

The following references have been added.

Ekman, L. et al. A shotgun metagenomic investigation of the microbiota of udder cleft dermatitis in comparison to healthy skin in dairy cows. *Plos one* **15**, e0242880 (2020).

Olson, N. D., Zook, J. M., Morrow, J. B. & Lin, N. J. Challenging a bioinformatic tool's ability to detect microbial contaminants using in silico whole genome sequencing data. *PeerJ* **5**, e3729 (2017).

Simon, H. Y., Siddle, K. J., Park, D. J. & Sabeti, P. C. Benchmarking metagenomics tools for taxonomic classification. *Cell* **178**, 779–794 (2019).

Tan, C. C. et al. No evidence for a common blood microbiome based on a population study of 9,770 healthy humans. **bioRxiv** (2022).

Chrisman, B. et al. The human “contaminome”: Bacterial, viral, and computational contamination in whole genome sequences from 1,000 families. *Sci. Reports* **12**, 9863 (2022).

The following references have been removed.

Oh, J. et al. Biogeography and individuality shape function in the human skin metagenome. **Nature** *514*, 59–64 (2014).

Mas-Lloret, J. et al. Gut microbiome diversity detected by high-coverage 16s and shotgun sequencing of paired stool and colon sample. *Sci. data* **7**, 1–13 (2020).

Multiple minor typo corrections were made (shown as tracked changes in the revised manuscript)

Reviewers' comments:

Reviewer #1 (Remarks to the Author):

In this revision, the authors have made changes to the manuscript that substantively addressed my previous concerns. I applaud the authors for tackling those concerns head-on, and I have no further major concerns.

Minor comments related to wording that don't have to be addressed:

"However, generating such experimental negative controls can be time consuming and expensive. Researchers have to perform extra experiments and do extra sequencing runs on empty samples to generate these controls. This extra work means that people must spend resources, including time and money, and as a result negative controls are often not generated."

I understand the point here, but the language feels too strong in that it might encourage microbiome practitioners to eschew negative controls in the future, rather than explaining why many did not include them in the past.

"Squeegee is the first de novo computational tool designed to identify and mark taxa as potential contaminants in the absence of "kit negative" or environmental contaminant controls."..."In summary, Squeegee is the first computational method for identifying potential microbial contaminants in the absence of environmental negative control samples."

Does decontam-frequency also fit this description? Squeegee doesn't require DNA concentrations though.

LEGEND

Author responses to editor/reviewers in **bold black**

Changes to the manuscript in **blue**

Location of inserted text in (parenthesis)

Added/updated display items indicated in **red**

Reviewers' comments in black

We thank the editor for providing the opportunity to submit a revised version of our manuscript. Although addressing the following comments related to wording are optional, we would still like to address those concerns and make some changes to our manuscript. Point-by-point responses to reviewer feedback are provided below.

R1Q1. "However, generating such experimental negative controls can be time consuming and expensive. Researchers have to perform extra experiments and do extra sequencing runs on empty samples to generate these controls. This extra work means that people must spend resources, including time and money, and as a result negative controls are often not generated." I understand the point here, but the language feels too strong in that it might encourage microbiome practitioners to eschew negative controls in the future, rather than explaining why many did not include them in the past.

We thank the reviewers for the feedback. The following text has been changed.

> (Introduction, Page 2)

However, the additional costs (both time and resources) to include negative control experiments are often a barrier to utilization. As a result, negative control experiments available for publicly available datasets are often lacking.

R1Q2. "Squeegee is the first *de novo* computational tool designed to identify and mark taxa as potential contaminants in the absence of "kit negative" or environmental contaminant controls."..." In summary, Squeegee is the first computational method for identifying potential microbial contaminants in the absence of environmental negative control samples." Does decontam-frequency also fit this description? Squeegee doesn't require DNA concentrations though.

Yes, running decontam with its frequency mode also fits this description. The following text has been changed to clarify that.

> (Discussion, Page 4)

Squeegee is the first *de novo* computational tool **specifically** designed to identify and **nominate** taxa as potential contaminants in the absence of "kit negative", **environmental contaminant controls, and other auxiliary data.**

> (Discussion, Page 6)

In summary, Squeegie is the first *de novo* computational method for identifying potential microbial contaminants in microbiome datasets in the absence of environmental negative control samples and auxiliary information such as DNA concentration information.

Other minor changes:

> (Figure 6)

Figure 6 has been re-plotted to align the left and right panel correctly, redundant tick labels have been removed, bold panel labels “a” and “b” have been added according to the figure formatting requirement. Exact p-values have been annotated according to the figure formatting requirement. No context in the figures has been changed.

> (Figure 1-6)

All figures in the main text have been re-generated in the vectorized pdf format. Parentheses in the panel labels have been removed according to the figure formatting requirement. No context in the figures has been changed.